# Microbiota-driven transcriptional changes in prefrontal cortex override genetic differences in social behavior

Mar Gacias[1]*, Sevasti Gaspari[1], Patricia-Mae G Santos[1], Sabrina Tamburini[2,3], Monica Andrade[2,3], Fan Zhang[1], Nan Shen[2,3], Vladimir Tolstikov[4], Michael A Kiebish[4], Jeffrey L Dupree[5], Venetia Zachariou[1,6], Jose C Clemente[2,3,7], Patrizia Casaccia[1,3]*

[1]Department of Neuroscience, Icahn School of Medicine at Mount Sinai, New York, United States; [2]Icahn Institute for Genomics and Multiscale Biology, Icahn School of Medicine at Mount Sinai, New York, United States; [3]Department of Genetics and Genomic Sciences, Icahn School of Medicine at Mount Sinai, New York, United States; [4]BERG, Framingham, United States; [5]Department of Anatomy and Neurobiology, Virginia Commonwealth University, Richmond, United States; [6]Department of Pharmacology and Systems Therapeutics, Icahn School of Medicine at Mount Sinai, New York, United States; [7]Immunology Institute, Icahn School of Medicine at Mount Sinai, New York, United States

*For correspondence: mar.gacias-monserrat@mssm.edu (MG); patrizia.casaccia@mssm.edu (PC)

Competing interests: The authors declare that no competing interests exist.

**Abstract** Gene-environment interactions impact the development of neuropsychiatric disorders, but the relative contributions are unclear. Here, we identify gut microbiota as sufficient to induce depressive-like behaviors in genetically distinct mouse strains. Daily gavage of vehicle (dH2O) in nonobese diabetic (NOD) mice induced a social avoidance behavior that was not observed in C57BL/6 mice. This was not observed in NOD animals with depleted microbiota via oral administration of antibiotics. Transfer of intestinal microbiota, including members of the Clostridiales, *Lachnospiraceae* and *Ruminococcaceae*, from vehicle-gavaged NOD donors to microbiota-depleted C57BL/6 recipients was sufficient to induce social avoidance and change gene expression and myelination in the prefrontal cortex. Metabolomic analysis identified increased cresol levels in these mice, and exposure of cultured oligodendrocytes to this metabolite prevented myelin gene expression and differentiation. Our results thus demonstrate that the gut microbiota modifies the synthesis of key metabolites affecting gene expression in the prefrontal cortex, thereby modulating social behavior.

## Introduction

Despite the diffuse prevalence of mental illness and the large efforts spent in identifying genetic elements of susceptibility, there is a need to define the role of environment—gene interactions. In addition to genetic predisposition, there is extensive epidemiologic literature emphasizing the role of environmental exposure in the development of mild to severe mood disorders. The aftermath of traumatic life events, for instance, is often characterized by the onset of severe depression or post-traumatic stress disorder (*Shalev et al., 1998*). The interplay between genes and environmental variables has gained recent attention, and several immunologic and lifestyle contributors have been proposed to modulate depressive symptoms. The detection of high levels of serum cytokines and the higher incidence of depression in patients with autoimmune disorders (*Postal and Appenzeller,*

**eLife digest** A combination of genes and environmental factors underlie an individual's risk of developing a mental illness. Among the environmental factors, it is becoming clear that communication between the gut and the brain is involved, but we do not understand how these two organs communicate. Our gut contains a variety of bacteria that help us to digest food and there is some evidence that changes in these bacterial communities can influence our mental health.

Transplanting feces from one individual to the gut of another is one way to alter the communities of bacteria in the gut. Here, Gacias et al. investigated whether fecal transplants are sufficient to induce social avoidance behavior – a symptom of depression – in mice. The experiments show that introducing specific combinations of bacteria into the gut is indeed able to cause healthy adult mice to avoid social interactions. This effect was caused by changes in the "myelin" sheath that surrounds many nerve fibers and could be prevented by giving the mice antibiotics, which decreased the number of bacteria in the gut.

Further experiments revealed that the mice that became depressed after fecal transplants had higher levels of a molecule called cresol, which is produced by certain gut bacteria. Gacias et al. found that cresol is able to reduce the amount of myelin produced by brain cells. Therefore, these findings show that changing the communities of bacteria in the gut can result in the accumulation of molecules that influence social behavior. Future work will aim to identify bacteria that can reduce the amount of cresol produced in the gut, which may have the potential to treat depression.

2015; Walker et al., 2011; Moll et al., 2011; van Hees et al., 2015; Feinstein et al., 2014) has suggested a role for neuroinflammation (Godbout et al., 2008; Menard et al., 2016; Audet et al., 2014). Deficiency of specific nutrients such as omega-3 fatty acids has been reported in subsets of patients with mental illnesses (Ohara, 2005; Patrick and Ames, 2015; Poudel-Tandukar et al., 2009; Panagiotakos et al., 2010), highlighting the link between mood disorders and the bioavailability of metabolites.

There is evidence that bioactive metabolites act as mediators of gut—brain communication, as shifts in gut microbial composition impact brain neurochemistry (Cryan and Dinan, 2012; Collins et al., 2012; Desbonnet et al., 2014; Bercik et al., 2010; 2011). Indeed, psychiatric comorbidities often accompany conditions characterized by an aberrant gut microbiota composition, such as irritable bowel syndrome, functional gastrointestinal disorder, and inflammatory bowel disease (Gevers et al., 2014; Morgan et al., 2012; Haberman et al., 2014; Carroll et al., 2011; Addolorato et al., 1997). Conversely, altered gut microbiota composition and function have been reported in patients with major depressive disorders and children with autism (Jiang et al., 2015; De Angelis et al., 2015; De Angelis et al., 2013; Parracho et al., 2005). The gut microbiota is a complex microbial ecosystem that rapidly responds to environmental changes and can modulate brain development, function, and behavior (Cryan and Dinan, 2012; Collins et al., 2012; Desbonnet et al., 2014; Bercik et al., 2010; 2011; Wu et al., 2011; Daniel et al., 2014; Lax et al., 2014). These studies suggest that social behavior may be affected by abnormal interactions between gut microbiota and the brain, though the underlying mechanisms remain only partially understood.

One hypothesis for the pathogenesis of depressive-like behaviors has been suggested through studies on social isolation in mice (Liu et al., 2012; 2016; Makinodan et al., 2012), which revealed a reduction of myelinated fibers in the prefrontal cortex (PFC), associated with changes in the oligodendrocyte transcriptome (Liu et al., 2012; 2016). Myelination is a dynamic process that continues into adulthood and contributes to physiologic brain function (Liu et al., 2012; 2016; Makinodan et al., 2012; Sánchez et al., 1998; Gibson et al., 2014; McKenzie et al., 2014). Oligodendrocytes are the myelinating cells of the central nervous system (CNS), and neuropathologic and transcriptomic studies have reported downregulated oligodendroglial transcripts and reduced myelin thickness in the brains of patients with schizophrenia, major depression, and bipolar disorder (Tkachev et al., 2003; Aston et al., 2005; Katsel et al., 2005). These data highlight the role of myelin in mental illness and depressive-like behaviors, though it remains to be established

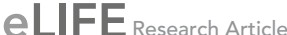

**Figure 1.** The strain-specific social avoidance behavioral response to daily gavage is affected by oral antibiotic treatment. (**A**) Experimental timeline: vehicle or antibiotic mix were administered daily by gastric gavage (g.g.) for 14 days. Behavioral testing was performed before (baseline) and after treatment. (**B–D**) Results of the Social Interaction (SI) test for NOD (**B**) and C57BL/6 (**D**) mice. Oral antibiotic treatment did not affect locomotor activity measured during the social interaction test (**C,E**) (3 independent experiments with 8 mice per group/experiment for a total of n=23–24 mice per condition). Data are mean ± S.E.M; *p<0.05, **p<0.01 based on one-way ANOVA with Bonferroni's post hoc test; n.s. indicates not significant.

*Figure 1 continued*

The following figure supplements are available for figure 1:

**Figure supplement 1.** The subcutaneous delivery of vehicle or antibiotic did not induce social avoidance behavior.

**Figure supplement 2.** Effect of subcutaneous or oral antibiotic treatment on body weight and macroscopic appearance of large intestine.

**Figure supplement 3.** Oral antibiotic treatment is well tolerated by recipients.

whether myelination in the adult PFC and social behavior are affected by alterations in gut microbiota composition. This study characterizes the gut microbiota in mice with social avoidance behavior and demonstrates that transfer of specific bacterial taxa is sufficient to alter adult PFC myelination and results in behavioral changes consistent with a depressive-like phenotype.

## Results

### Non-obese diabetic (NOD) and C57BL/6 mice display differential susceptibility to develop depressive-like symptoms in response to daily gavage

Although gastric gavage and subcutaneous injections are routine, daily procedures used to administer drugs or special diet to rodents, the potential behavioral effects they may induce in mice have not been investigated. Daily gastric gavage with vehicle for two weeks (*Figure 1A*) was sufficient to induce social avoidance behavior in NOD mice (*Figure 1B*), without affecting their overall locomotor activity (*Figure 1C*). This depressive-like behavior induced by daily gavage was dependent on the specific mouse strain, as C57BL/6 mice were not affected (*Figure 1D,E*) (*Moy et al., 2008*). Subcutaneous injection of vehicle did not elicit any behavioral effect in either strain (*Figure 1—figure supplement 1*). Daily gastric gavage with an antibiotic cocktail proven to deplete the gut microbiota (*Reikvam et al., 2011*) failed to induce the social avoidance behavior in NOD mice (*Figure 1B*), and similarly had no effect on the C57BL/6 mice (*Figure 1D*). The antibiotic regimen was well tolerated by both NOD and C57BL/6 mice, did not impact body weight or glucose levels, and did not result in any gastric hemorrhage or visible stomach damage (*Figure 1—figure supplement 2* and *Figure 1—figure supplement 3*). Consistent with previous reports, only chronic oral antibiotic treatment (but not subcutaneous delivery) induced enlargement of the large intestine (*Figure 1—figure supplement 2*), a macroscopic sign associated with microbiota depletion (*Reikvam et al., 2011*). Interestingly, daily gavage also induced an anxiety-like behavior in both NOD and C57BL/6 mice, as revealed by the elevated plus maze (EPM) (*Figure 2B,D*). However, the anxiety-like behavioral change displayed in response to daily gavage was not affected by oral antibiotic treatment (*Figure 2B,D*), suggesting that only the depressive-like behavior is mediated by alterations in gut microbiota. To further validate this hypothesis, we conducted the forced swim test (FST), which is considered a measure of despair-like behavior, in NOD and C57BL/6 mice after daily gavage with either vehicle or antibiotics. The despair-like behavior was induced by vehicle gavage in the NOD strain, and was prevented by oral antibiotic treatment (*Figure 2C*), but was not detected in the C57BL/6 mice (*Figure 2E*). Together, these results indicate that daily gavage of vehicle induces social avoidance and despair-like behaviors in NOD mice, but not in C57BL/6 mice, and that this effect is not observed when gavaging antibiotics orally and not subcutaneously.

### Behavioral differences in genetically different mice are associated with specific gut microbiota composition

To further characterize the effect of vehicle and antibiotic treatment on gut microbiota composition, we conducted 16S rRNA sequencing analysis of cecal and fecal samples collected after behavioral testing and after 14 days of treatment (*Figure 3A*). Unweighted UniFrac distances (*Lozupone and Knight, 2005*) were calculated between all pairs of fecal samples based on their microbiota

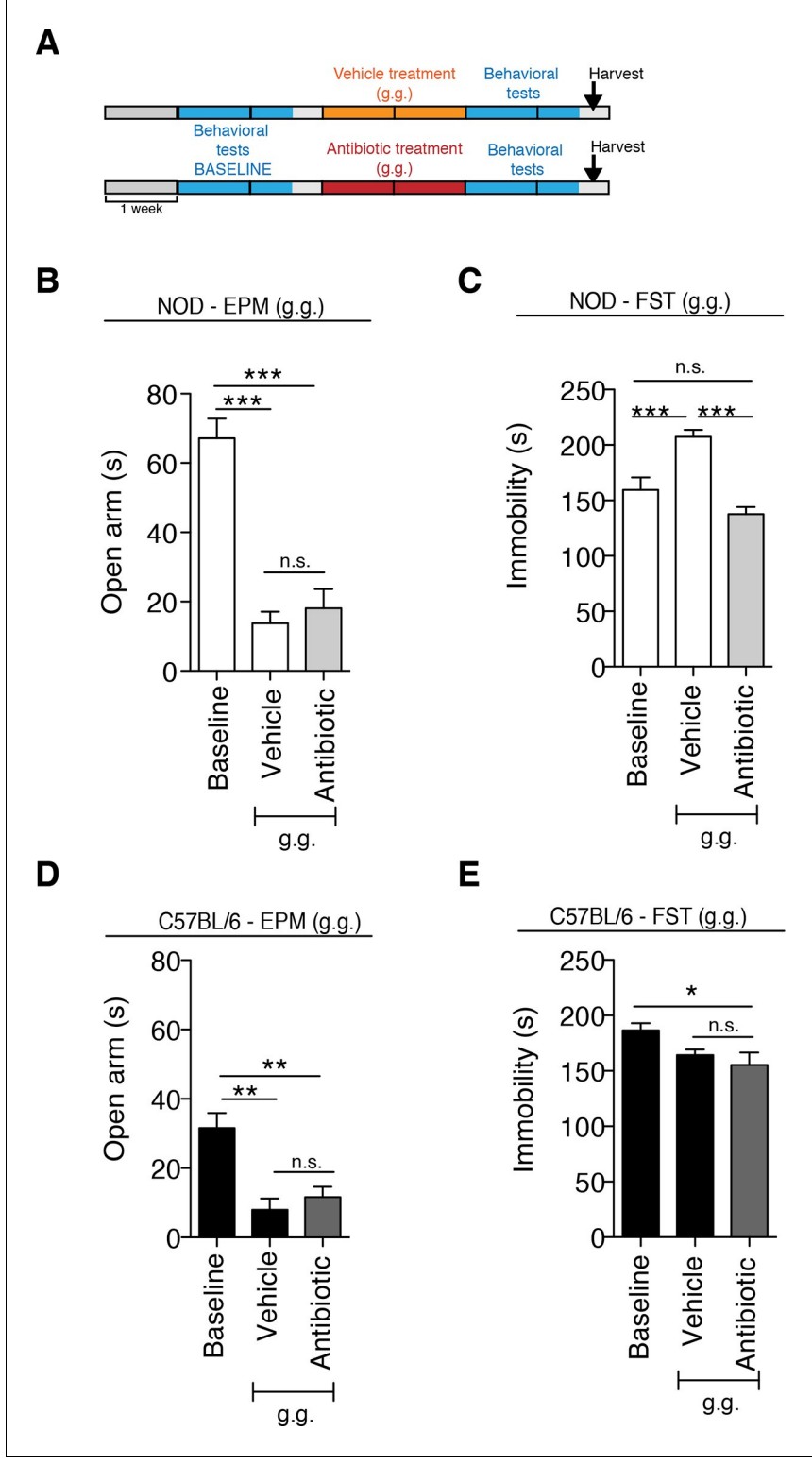

**Figure 2.** The strain-specific anxiety- and despair-like behavioral responses to daily gavage are differentially affected by oral antibiotic treatment. (**A**) Experimental timeline: vehicle or antibiotic mix were administered daily by gastric gavage (g.g.) for 14 days. Behavioral testing was performed before (baseline) and after treatment. Figure shows the results for the Elevated Plus Maze (EPM) and Forced Swim Test (FST) for NOD (**B**, **C**) and C57BL/ 6 (**D**, **E**) mice after oral treatment (g.g.). Baseline measurements for FST were performed in a separate cohort of mice (*n*=10) to avoid carryover effects (3 independent experiments with 8 mice per group/experiment for a total of

*Figure 2 continued on next page*

*Figure 2 continued*

*n*=24 mice per condition). Data are mean ± S.E.M; *p<0.05, **p<0.01, ***p<0.001 based on one-way ANOVA followed by Bonferroni's post hoc test; n.s. indicates not significant.

The following figure supplement is available for figure 2:

**Figure supplement 1.** Anxiety and despair-like behaviors after subcutaneous (s.c.) vehicle or antibiotic treatment.

composition. Based on these distances, Principal coordinate analysis (PCoA), an ordination method conceptually similar to principal component analysis, revealed a clear separation between vehicle-gavaged and baseline NOD (*Figure 3*) and between vehicle and antibiotic treated NOD and C57BL/6 mice (*Figure 3—figure supplement 1A*). PCoA analysis revealed clear differences between NOD mice before ("baseline") and after oral treatment with antibiotics (*Figure 3B*), with differences also observed between samples before and after treatment with vehicle (*Figure 3C*). Since the depressive-like behavior was only observed in oral vehicle-treated NOD mice, we focused on identifying the specific microbiota that differ in these animals before and after treatment. Analysis of Operational Taxonomic Units (OTUs, defined as groups of 16S rRNA gene sequences with high similarity and that broadly correspond to a bacterial species) identified several taxonomic groups that were exclusively found in the vehicle-treated mice (*Figure 3D* and *Gacias et al., 2016*). These taxa represent potential candidates associated with the depressive-like phenotype observed in NOD mice. Linear discriminant analysis effect size (LEfSe) (*Segata et al., 2011*), a biomarker discovery method based on the Kruskal–Wallis and Wilcoxon tests, was used to identify key bacterial taxa enriched in vehicle-treated versus antibiotic-treated animals in each strain (*Figure 3—figure supplement 1B–E*). As expected, Proteobacteria were enriched in antibiotic-treated animals, while vehicle-treated mice had enrichment in Bacteroidetes and Firmicutes (*Figure 3—figure supplement 1C,E*).

Analysis of tissue samples revealed similar differences between vehicle- and antibiotic-treated mice in both strains (p<0.01, adonis with 999 permutations). No significant changes in gut microbiota composition were detected when antibiotics were administered subcutaneously.

## Modification of the gut microbiota following oral antibiotic administration induces unique changes in the medial prefrontal cortex adult myelination of NOD mice

To identify possible CNS transcriptional signatures associated with the behavioral outcomes described in the vehicle-gavaged, but not antibiotic-gavaged, NOD mice, we performed an unbiased transcriptomic analysis of the medial prefrontal cortex (mPFC) using RNA sequencing. This analysis revealed decreased expression of genes related to myelination (*Figure 4* and *Figure 4—figure supplements 1,2* and *Gacias et al., 2016*) in vehicle-gavaged NOD mice - characterized by social avoidance behavior - compared to antibiotic-treated mice, whose behavior was comparable to baseline controls (*Figure 1*). The differences in myelin gene transcripts in the mPFC of vehicle-gavaged NOD compared to antibiotic-treated mice were validated by quantitative real-time qPCR (*Figure 4B*) and immunohistochemistry (*Figure 4C*). These differences were detected only in NOD mice, and not in C57BL/6 mice that showed no change in social behavior with oral gavage (*Figure 4D,E*). The differences in myelin gene expression in the mPFC could not be attributed to a nonspecific effect of antibiotic treatment, as there were no differences observed after subcutaneous delivery (*Figure 4—figure supplement 1*). The regional specificity of the transcriptional changes was also assessed in NOD mice by evaluating samples from a distinct brain region, the nucleus accumbens (NAc), revealing no difference in the two treatment groups (*Figure 4—figure supplement 1*). These data provide further support for the relationship between defective mPFC adult myelination and depressive-like behavior, as indicated by the lower levels of myelin transcripts and reduced area of MBP immunostaining in vehicle-gavaged NOD mice exhibiting social avoidance. The results also demonstrate that the transcriptional and behavioral effects were prevented by oral antibiotic treatment.

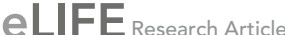

**Figure 3.** Enrichment of bacterial OTUs induced by gastric gavage (g.g.) in NOD mice. (**A**) Experimental timeline indicating time points of fecal collection (arrows) relative to behavioral testing and treatment. (**B,C**) Principal coordinate analysis plots of unweighted UniFrac distances of microbiota in fecal samples at baseline and after 14 days of daily g.g. of antibiotics or vehicle in NOD mice. Each dot represents the microbiota of a sample (1 sample = feces pooled from 3–5 mice), color-coded by treatment (vehicle or antibiotic) and time-point. The percentage of variation explained by each principal coordinate (PC) is shown in parentheses. All samples were rarefied at 5000 sequences. (**D**) Analysis of unique Operational Taxonomic Units (OTUs) present in NOD vehicle-treated mice

*Figure 3 continued on next page*

*Figure 3 continued*

compared to their fecal microbiota at baseline. Figure shows representative taxa enriched in fecal samples of NOD vehicle-treated mice compared to their baseline samples. Each bar represents the microbiota of an individual sample (1 sample = 3–5 mice per cage). See *Gacias et al. (2016)*.

The following figure supplement is available for figure 3:

**Figure supplement 1.** Oral antibiotic treatment effectively modifies the microbiota composition in NOD and C57BL/6 mice.

## Transplantation of fecal microbiota from vehicle-treated NOD mice to depleted C57BL/6 recipients is sufficient to recolonize the gut and transfer transcriptional, and behavioral traits

To determine whether the social avoidance behavior and mPFC transcriptional changes induced by daily gavage of vehicle in NOD mice were caused by the enrichment of specific gut bacteria, we transferred the cecal content of vehicle-treated or antibiotic-treated NOD mice into C57BL/6 recipients, whose endogenous flora had been depleted by antibiotic treatment (*Figure 5A*). Social behavior in C57BL/6 depleted recipients was assessed before and after transplantation with microbiota from either vehicle-gavaged (Group I) or antibiotic-gavaged (Group II) NOD donors. The behavior of the C57BL/6 recipients resembled that of the donors: Social avoidance behavior was detected in Group I recipients, and was not observed in Group II recipients (*Figure 5B,C*). Intriguingly, transplantation of vehicle-gavaged NOD microbiota also transferred the transcriptional changes in the mPFC, but not in the NAc, as shown by the lower levels of myelin gene transcripts (*Mag, Mog, Plp1, Mobp*) in Group I mice compared to Group II recipients (*Figure 5D*). The functional consequences of the transcriptional changes in myelin genes were further validated by electron microscopy, and ultra-structural analysis revealed decreased myelin thickness in Group I recipients displaying a social avoidance behavior (*Figure 5E*). Quantification of myelin thickness relative to axonal diameter (*g* ratio) revealed that Group I recipients transplanted with vehicle-gavaged NOD microbiota, presented thinner myelin than Group II, recipients of antibiotic-treated NOD donors. No significant differences between the two groups were observed in the NAc (*Figure 5E*). The transfer of depressive-like behavior from donor to recipient was further validated by the detection of increased immobility at the FST in Group I mice compared to Group II (*Figure 5—figure supplement 1*).

Collectively, these findings suggest that the gut microbiota of vehicle-gavaged NOD donors was sufficient to transfer the depressive-like behavior, modulate transcript levels in the mPFC, and impact region-specific adult myelination in microbiota-depleted C57BL/6 recipients.

The genomic DNA content was measured in fecal pellets of C57BL/6 recipients to validate the depletion of the gut microbiota with 14 days of antibiotic treatment, and to evaluate the effectiveness of recolonization after transplantation (*Figure 6B,D*). Analysis of alpha diversity (the number of bacterial taxa present in a sample or group of samples) further confirmed the microbiota depletion (*Figure 6C,E*). In both groups, diversity was significantly reduced from baseline after antibiotic treatment (*Figure 6C,E*; p<0.01 ANOVA with Tukey's honest significant difference (HSD) post-hoc analysis). As expected, after transplantation Group II mice still exhibited a significantly depleted diversity compared to baseline (*Figure 6E*; p<0.01 ANOVA with Tukey's HSD), while bacterial diversity in Group I had recovered to levels similar to baseline and was not significantly different (*Figure 6C*; p=0.09, ANOVA with Tukey's HSD). These results suggested that transfer of behavioral traits was associated with restoration of bacterial diversity to baseline levels. In order to determine the differences in microbiota compositions associated with the behavioral phenotype, we conducted PCoA analysis based on unweighted UniFrac analysis (*Figure 6F*). Although all pooled fecal samples from NOD donors and C57BL/6 recipients clustered together at baseline (samples on the right side of the plot), treatment with antibiotics resulted in a drastic reshaping of the bacterial communities of both NOD (middle of the plot) and C57BL/6 (bottom-left side) mice. The microbiota composition of Group II mice after transplant (which did not display social avoidance behavior) was distinct from baseline, similar to antibiotic-treated animals pre-transplant (top-left side). However, Group I recipients which displayed social avoidance behavior (#19 and #18 on the plot), had compositions that were close to those of their vehicle-treated NOD donors. In contrast, Group I recipients which did

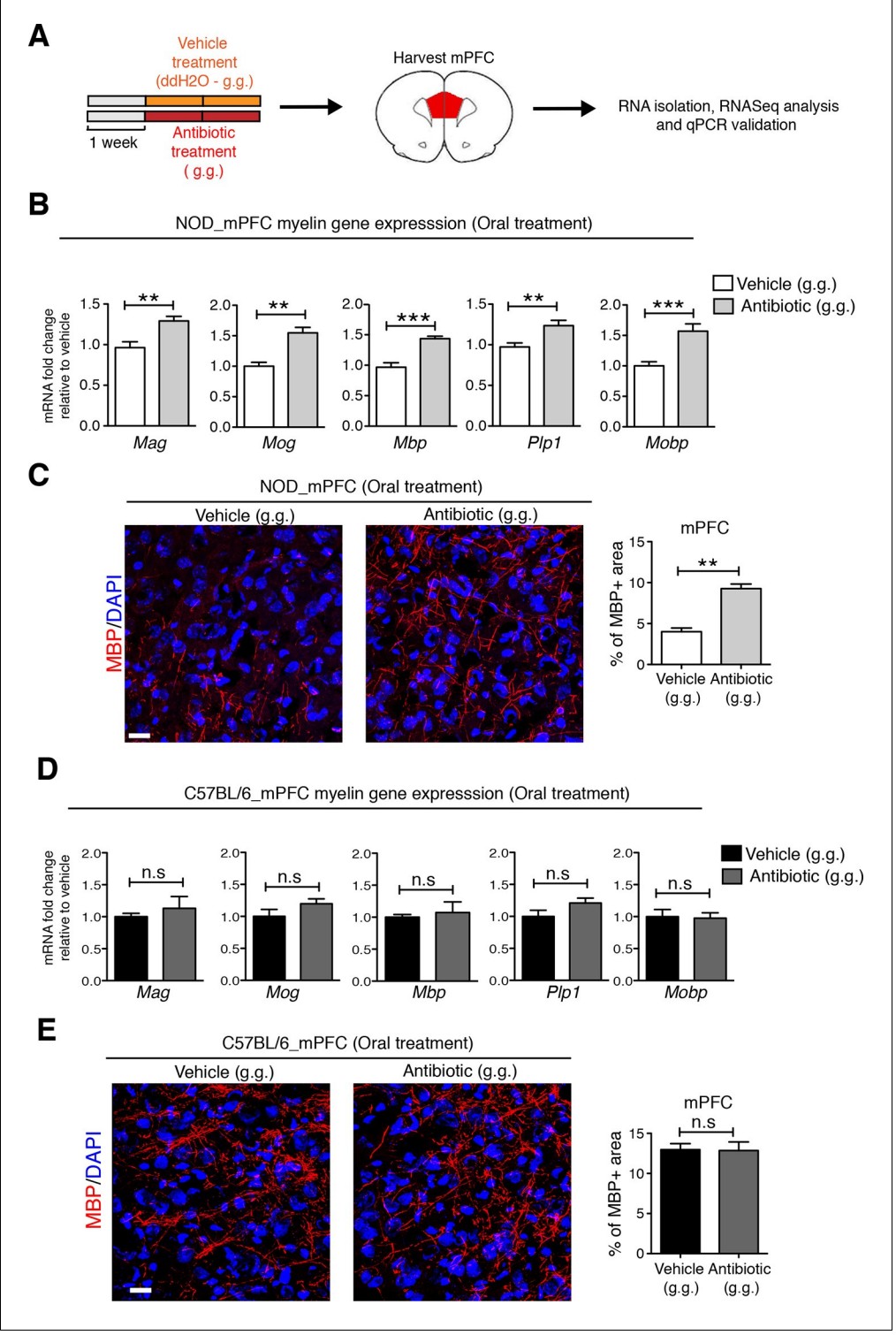

**Figure 4.** Myelin transcripts and myelinated fibers in the medial prefrontal cortex (mPFC) of adult NOD mice with social avoidance behavior. (**A**) Experimental timeline: vehicle or antibiotic mix were administered daily by gastric gavage (g.g.) for 14 days and mPFC was harvested for RNA extraction and quantitative real-time qPCR or immunohistochemsitry (**B,D**) qPCR of myelin transcripts after vehicle or antibiotic treatment of NOD (**B**) and C57BL/6 (**D**) mice. Values were normalized to *36b4* mRNA levels and are referred as fold change relative to vehicle-treated values (*n*=6 mice per group). (**C,E**) Representative confocal images and quantification of MBP+ fibers (red) in mPFC of NOD (**C**) and C57BL/6 (**E**) mice after vehicle or antibiotic treatment. DAPI (blue) was used
*Figure 4 continued on next page*

*Figure 4 continued*

as nuclear counterstain. Scale bar, 50 μm. Graph represents quantification of MBP+ fibers per surface area (*n*=3 for NOD; *n*=4 for C57BL/6). Data are mean ± S.E.M; **p<0.01, ***p<0.001 based on unpaired *t* test. n.s. indicates not significant.

The following figure supplements are available for figure 4:

**Figure supplement 1.** Regional specificity of myelin changes in response to antibiotic treatment.

**Figure supplement 2.** Effect of oral antibiotic treatment on the transcriptional profile in medial prefrontal cortex (mPFC).

not display social avoidance behavior (#17 on the plot), clustered with Group II recipients. This result suggests that the transplant procedure was not equally effective in all Group I mice. The distance in microbiota composition between vehicle-gavaged donors and recipients was significantly correlated with the social avoidance behavior, as measured by social interaction time (*Figure 6G*; p=0.01). This result suggests that the ability to successfully transfer the gut microbiota from vehicle-gavaged NOD donors was significantly correlated with the transmission of the depressive-like behavior. LEfSe analysis revealed a number of taxa that were significantly different between Group I and Group II C57BL/6 recipients (*Figure 6—figure supplement 1* and *Gacias et al., 2016*). We further refined this analysis, by identifying the specific OTUs transferred from vehicle-gavaged NOD donors to Group I recipients (*Figure 6—figure supplement 2* and *Gacias et al., 2016*). Members of the Clostridiales order, including *Lachnospiraceae* and *Ruminococcaceae*, were among those present in equal proportions both in the donors and the recipients in Group I recipients displaying a depressive-like behavior (i.e. samples #18 and #19), while absent in Group I recipients that did not exhibit such behavior (i.e. sample#17; *Figure 6F* and *Gacias et al., 2016*). We further confirmed these taxa as potentially responsible for this phenotype by qPCR using primers specific to these bacterial groups (*Figure 6—figure supplement 2C*). In order to identify differences undetectable at the OTU level, we performed oligotype analysis in those OTUs established as potentially responsible for the depressive-like behavior (*Segata et al., 2011*; *Eren et al., 2014*). Oligotype analysis is an entropy-based method to identify single nucleotide differences in sequences from closely related organisms. We found that most OTUs were composed of a single high-abundance oligotype (*Figure 6—figure supplement 3A–C,E, G–O*) and therefore support the conclusions from the OTU-level analysis. However, we identified three OTUs that had two oligotypes with similar abundances and distribution across samples: OTU 183849 (*Blautia producta*, a member of the Lachnospiraceae, *Figure 6—figure supplement 3D*), 188840 (unidentified member within *Lachnospiraceae*, *Figure 6—figure supplement 3F*), and 4418586 (unidentified member within Clostridiales, *Figure 6—figure supplement 3P*). Additional inspection of these sequences revealed the oligotypes GTT and TTT from the *Blautia producta* OTU, as well as the TG and TT oligotypes from the *Lachnospiraceae* OTU, had *B. producta* JCM 1471 as the closest reference sequence in NCBI; the oligotypes from Clostridiales had no close reference sequence. Overall, these results show that either a single oligotype or a combination of two oligotypes with similar abundance distributions were dominant within the analyzed OTUs, which suggested they might drive the observed social phenotypes.

The gut metabolome is altered in microbiota-transplanted C57BL/6 mice displaying altered social and despair-like behaviors.

Several studies have demonstrated that gut metabolites can impact the homeostatic host-microbiota interactions and affect behavior (*Daniel et al., 2014*; *Hsiao et al., 2013*). To determine whether altered taxa in the gut microbiota could impact the levels of metabolites, which in turn drive behavioral and transcriptional changes observed in the mPFC, we performed an unbiased metabolomic analysis of gut tissue from C57BL/6 recipients with (Group I) and without (Group II) social avoidance behavior (*Figure 7*). The analysis included non-targeted and targeted protocols and gas chromatography combined with time-of-flight high-resolution mass spectrometry, hydrophilic liquid chromatography coupled with high-resolution mass spectrometry and hydrophilic interaction chromatography with liquid chromatography and tandem mass-spectrometry for the study of monoamine to neurotransmitters (*Tolstikov et al., 2014*; *Danaceau et al., 2012*). After statistical

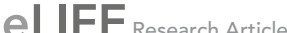

**Figure 5.** Social avoidance behavior transfer from NOD donors to microbiota depleted C57BL/6 by fecal transplantation. (**A**) Experimental timeline for donor (NOD) and transplant-recipient (C57BL/6) mice. (**B**,**C**) Results from Social Interaction (SI) tests conducted in C57BL/6 recipients before and after transplantation with either microbiota from vehicle-treated (Group I; **B**) or antibiotic-treated (Group II, **C**) NOD mice. Graphs represent the amount of time spent (seconds) in the interaction zone when a target is present. Red dashed bar represents the interaction time of the NOD donors. Data are mean ± S.E.M; *p<0.05, **p<0.01 based on a two-way ANOVA

*Figure 5 continued on next page*

*Figure 5 continued*

(*n*=12 mice per experiment, 2 replicates of 12 for a total of 24 mice per condition). (D) Graphs indicate the relative levels of myelin gene transcripts in mPFC and NAc of C57BL/6 recipients displaying (Group I) or not displaying (Group II) social avoidance behavior after transplantation with NOD microbiota (*n*=6–8 mice per group; *p<0.05, **p<0.01, ***p<0.001 based on unpaired *t* test). (E) Electron micrographs and quantified *g*-ratios of myelinated axons in mPFC and NAc in Group I and Group II C57BL/6 recipients after transplantation with the NOD microbiota. Scale bar, 1 μm. (*n*=3 per treatment and condition; statistical differences between groups were determined using two-tailed *t*-test; n.s. indicates not significant).
The following figure supplement is available for figure 5:

**Figure supplement 1.** Effect of NOD vehicle-treated microbiota on the despair-like behavior of C57BL/6 recipients.

corrections and normalization, we conducted Partial Least Squares-Discriminant Analysis (PLS-DA), a method that incorporates elements from principal component analysis, regression, and linear discriminant analysis, which revealed a clear separation of the overall gut metabolites between Group I and Group II (*Figure 7B*). A total of 382 metabolites were detected in the guts of C57BL/6 transplant recipients (*Gacias et al., 2016*) A first pathway impact analysis provided a visual representation of the most dramatically affected pathways between the two groups, and identified the linoleic/linolenic acid and phenylalanine/tryptophan synthetic pathways as differentially represented in the two sets of samples (*Figure 7C*, *Table 1* and *Gacias et al., 2016*). Further evaluation of the metabolome using a volcano plot representing individual differences in metabolites revealed increased levels of cresol, stearamide, N-acetylasparagine, and oleamide in Group I recipients, which displayed social avoidance and despair-like behaviors (*Figure 7D,E*). As cresol is a highly permeable compound that was detected at high levels in the guts of Group I mice characterized by behavioral changes and impaired mPFC myelination, we treated primary cultured oligodendrocyte progenitors with increasing concentrations of cresol and tested for myelin gene expression (*Figure 8*). Expression of *Mag, Mog, Mbp*, and *Cnp* transcripts and the number of double-positive CNP+/OLIG2+ cells were reduced incresol-treated cultures compared to controls (*Figure 8A–C*). However, this effect was not due to toxicity, but rather to impaired differentiation, as indicated by the increased transcripts of immature progenitor markers (*Pdgfra*) and the stable OLIG2+ cell counts (*Figure 8D,E*).

## Discussion

Our results provide strong evidence that manipulations of gut microbiota are sufficient to induce depressive-like behaviors in adult mice. The behavioral changes were detected in mice with gut microbiota enriched for the taxa Clostridiales, including the *Lachnospiraceae* and *Ruminococcaceae* families, and with increased levels of highly permeable metabolites (such as cresol) with the ability to impair oligodendrocyte differentiation and myelin gene transcription. The observation that behavioral traits were only detected in transplant recipients with effective colonization of these taxa highlights the potential molecular mechanisms by which gut microbiota impacts CNS homeostasis.

To date, several studies have focused on the relationship between microbiota composition and the development of anxiety-like behaviors. Dysbiotic microbiota induced by either pathogenic infections or antibiotic treatment has been shown to increase anxiety-like behavior in conventionally raised mice (*Bercik et al., 2010*; *2011*; *Lyte et al., 2006*), while germ-free mice show reduced levels of anxiety-like behaviors compared to normal mice (*Diaz Heijtz et al., 2011*; *Neufeld et al., 2011*). In our study, anxiety-like behaviors were also shown to be affected by daily gastric manipulations affecting microbiota composition. However, social avoidance and despair-like behaviors were differentially induced by gavage in two genetically distinct strains of mice, which could be prevented by the administration of a broad-spectrum antibiotic cocktail. Daily gavage of NOD mice induced significant changes of gut bacterial communities and depressive-like behavior (social and despair-like behaviors), which was associated with enrichment of bacteria within the Clostridiales. Antibiotic treatment decreased the overall bacterial diversity and prevented the behavioral effects. Subcutaneous administration of the same antibiotic treatment failed to induce significant changes either in the

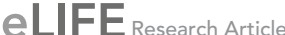

**Figure 6.** Effect of fecal transplantation on bacterial mass and biodiversity in microbiota depleted C57BL/6 recipients. (A) Experimental timeline for donors (NOD) and transplanted recipients (C57BL/6). (B,C) Graphs represent fecal biomass (μg of gDNA relative to total fecal weight) of C57BL/6 recipients prior to transplantation (#1 before and #2 after 14 days of antibiotic treatment) and at end point after-transplantation (#3) with donor microbiota (n=3 pooled samples per time-point, each sample represents 1 sample = pooled feces from 3–5 mice. Data are mean ± S.E.M; *p<0.05, **p<0.01 based on one-way ANOVA with Bonferroni's post hoc test). (C,E)

*Figure 6 continued on next page*

*Figure 6 continued*

Rarefaction curves comparing alpha diversity of fecal microbiota samples from C57BL/6 recipients at different experimental time-points (#1, #2, and #3). (F) Principal coordinate analysis plot of unweighted UniFrac distances of fecal samples from NOD donors and C57BL/6 mice at different time-points. (#1, #2, and #3). Each dot represents the microbiota of a sample, colored by group, treatment, and time-point (*n*=3 pooled samples per time-point; each sample corresponds to pooled feces from 3–5 mice). The percentage of variation explained by each principal coordinate (PC) is shown in parentheses. (E) Relationship between social interaction time and unweighted UniFrac distance to NOD donor mice (*n*=3) for all C57BL/6 recipients (*n*=10). Each point represents a single C57BL/6 animal, colored by group (light blue: Group_I, transplanted with NOD-vehicle microbiota; pink: Group_II, transplanted with NOD-antibiotic microbiota). Linear regression analysis indicates a significant correlation (p=0.0103) between the variables.

The following figure supplements are available for figure 6:

**Figure supplement 1.** Transfer of social avoidance behavior is associated with altered colonic composition of the microbiota.

**Figure supplement 2.** Social avoidance behavior is associated with enrichment of specific OTUs.

**Figure supplement 3.** Oligotype analysis of gut tissue samples.

microbiota composition or behavior, further highlighting the importance of a local effect of oral antibiotic treatment on these intestinal microbial communities.

To prove causality and understand whether behavioral changes observed in vehicle-gavaged NOD mice were in fact modulated by the intestinal microbiota, we transferred the cecal content of these mice (displaying a social avoidance behavior) or antibiotic-treated (with normal social behavior) NOD donors into the microbiota-depleted guts of C57BL/6 recipients. Our results demonstrate that only recipients with successful recolonization of the taxa enriched in the vehicle-treated NOD mice (e.g. Clostridiales, *Lachnospiraceae* and *Ruminococcaceae*) exhibited the social avoidance and despair-like behaviors, as well as the myelin gene expression in the mPFC of the donors. These transcriptional changes resulted in decreased adult PFC myelination in mice with transferred behavior. The microbiota of these C57BL/6 recipients showed significant differences in the abundance of several of the bacterial populations identified in the donors. Interestingly, alterations of some *Lachnospiraceae* and *Ruminococcaceae* spp. have been associated with behavioral deficits in mice (***Bruce-Keller et al., 2015***). Our results did not identify a single bacterium responsible for the behavioral changes induced by vehicle-gavage in the NOD mice or by transplantation in the C57Bl6 animals, suggesting that specific communities enriched in taxa from the *Lachnospiraceae* and *Ruminococcaceae* are responsible for the observed phenotype. Community-driven effects have also been reported in the induction of colonic regulatory T cells by specific mixtures of Clostridia strains in models of colitis or in cognitive and stereotypic behavioral changes induced by high-fat diet microbiota in non-obese mice (***Bruce-Keller et al., 2015***; ***Atarashi et al., 2015***).

Our results also show that alterations of the microbial composition modified gut-produced metabolites and transcriptomic profiles in the mPFC, subsequently affecting behavior (***Daniel et al., 2014***; ***Hsiao et al., 2013***). Microbiota composition has previously been shown to modulate anxiety-like behaviors in adult mice via changes in levels of brain-derived neurotrophic factor in the hippocampus (***Bercik et al., 2011***). The results of our untargeted transcriptomic analysis of the mPFC, the region responsible for the integration of external stimuli and complex behaviors (***Regenold et al., 2006***), identified a signature characterized by genes regulating transcription, circadian rhythm, protein phosphorylation, synapses, and myelin. Altered expression of genes related to myelin and circadian rhythm is consistent with reported white matter changes and sleep disruption in human patients with major depression (***Liu et al., 2015***; ***Landgraf et al., 2014***; ***Lavebratt et al., 2010***; ***Kishi et al., 2009***) as well as with the reported behavioral changes detected on myelin mutant mice (***Hagemeyer et al., 2012***). The association of diminished myelination in mPFC with the observed social avoidance behavior is supported by recent studies describing decreased myelin gene expression and fewer myelinated fibers in the mPFC of mice after prolonged social isolation (***Liu et al., 2012***; ***2016***; ***Makinodan et al., 2012***). Importantly, adoptive transfer of gut microbiota from NOD

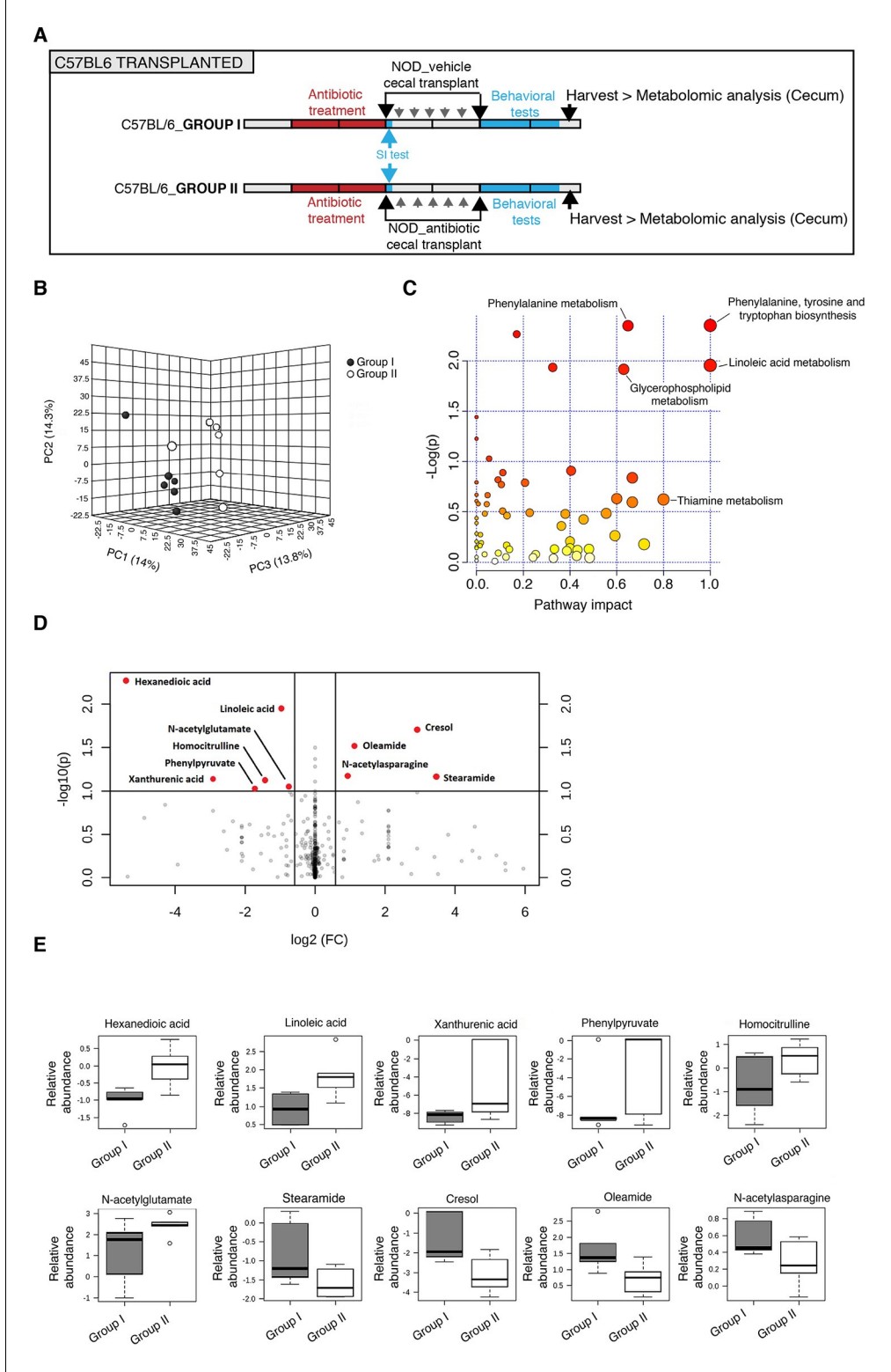

**Figure 7.** Metabolomic analysis of gut tissue from microbiota-transplanted C57BL/6 mice. (**A**) Experimental timeline. (**B**) 3D plot of scores between selected components generated by PLS-DA analysis comparing Group I (transplanted with microbiota from vehicle-treated NOD mice; filled circles) and Group II (transplanted with microbiota from antibiotic-treated NOD mice; open circles). (**C**) Metabolic pathway impact overview generated with MetaboAnalyst 3.0. Unaltered pathways have a score of 0, and the most impacted pathways have higher scores. Pathways having the least statistical significance score are uncolored, whereas pathways having a high

*Figure 7 continued on next page*

statistical significance score are colored in red. See *Gacias et al. (2016)*. (**D**) Metabolites with the greatest differential between mice with (Group I) and without (Group II) behavioral phenotype, were selected by volcano plot with a fold-change threshold of 1.5 (x axis) and t test threshold of 0.1 (y axis). Red circles represent metabolites above the threshold (Group II *vs* Group I); see *Table 1*. (**E**) One-way analysis of variance box and whisker plots illustrating the metabolite changes observed in Groups I and II. The y axis illustrates normalized, log transformed, and scaled peak area. Horizontal lines within the boxes represent the group means. Open circles represent excluded levels (outliers) (*n*=6 mice per group).

mice was able to recapitulate the mPFC transcriptional changes detected in recipient mice, thereby directly implicating gut microbiota as a causal factor for the induced behavioral and transcriptional changes.

One mechanism by which the gut microbiota may regulate such alterations is through the production of selective metabolites. Several recent studies have shown that a dysbiotic gut microbiota can produce neurotoxic metabolites directly impacting behavior (*Hsiao et al., 2013*; *Persico and Napolioni, 2013*; *Shaw, 2010*). For instance, in a mouse model for autism spectrum disorders during development, characterized by dysregulation of *Lachnospiraceae, Ruminococcaceae* the anxiety-like phenotype correlated with the levels of the metabolite 4-ethylphenylsulfate (4-EPS) (*Hsiao et al., 2013*). In our study, social avoidance behavior in adult mice was significantly associated with enrichment in *Lachnospiraceae, Ruminococcaceae* and Clostridiales and thedetection of high levels of cresol. This highly permeable metabolite was detected only in the gut of mice with social avoidance behavior, and was capable of preventing myelin gene expression and differentiation of oligodendrocyte progenitors into myelin-forming cells. These results suggest a potential mechanism linking CNS transcriptional changes to gut microbial homeostasis. Thereby increased intestinal production of

**Table 1.** Summary of trends in levels of cecal metabolites in C57BL/6 transplanted mice (Group II *vs* Group I).

| Super Pathway | Sub-pathway | Metabolite | Fold change (Group II vs I) | p value |
|---|---|---|---|---|
| Amino acid | Phenylalanine metabolism | Benzoic Acid | 1.01 | 0.031786 |
| Amino acid | Alanine, aspartate and Glutamate metabolism | N-acetylasparagine | 0.52 | 0.066953 |
| Amino acid | Tryptophan metabolism | Xanthurenic acid | 7.55 | 0.072748 |
| Amino acid | Urea cycle | Homocitrulline | 2.7 | 0.075261 |
| Amino acid | Arginine and proline metabolism | N-acetyl-glutamate | 1.68 | 0.08886 |
| Amino acid | Phenylalanine metabolism | phenylpyruvate | 3.3 | 0.094054 |
| Carbohydrate | Pentose phosphate pathway | Sedoheptulose-7-phosphate | 0.99 | 0.05242 |
| Cofactors and vitamins | Microbial metabolism in diverse environments | cresol | 0.13 | 0.019692 |
| Lipid | Fatty acids | Hexanedioic acid | 42.32 | 0.0053711 |
| Lipid | Long chain fatty acid | Linoleic acid | 1.95 | 0.011101 |
| Lipid | Long chain fatty amide | Oleamide | 0.46 | 0.030305 |
| Lipid | Long chain fatty acid | dihydroxystearic acid | 1.02 | 0.042845 |
| Lipid | Long chain fatty amide | Stearamide | 0.09 | 0.068105 |
| Nucleotide | Purine metabolism | cAMP | 0.99 | 0.077892 |

Data were analyzed using comprehensive global mass spectrometry-based metabolomics analysis. Additional details are provided in Experimental Procedures.

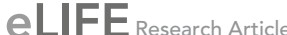

**Figure 8.** Cresol treatment decreases myelin gene expression. (**A**) Transcript levels of oligodendrocyte lineage (*Olig2*), progenitor (*Pdgfrα, Cspg4*) and differentiation (*Mag, Mog, Mbp, Cnp, Sox10*) markers in oligodendrocyte progenitors cultured in differentiating conditions and treated with increasing concentrations of cresol (0, 10, 50 µM). DMSO was used as vehicle and negative control. Values were normalized to *36b4* mRNA levels and are referred as fold change relative to the control group (*n*=3 independent primary cultures). (**B,C**) Representative confocal images and quantification of early differentiated oligodendrocytes (CNP+/OLIG2+) after treatment with increasing concentrations of cresol (0, 10, 50 µM) for 24 hr. (**D,E**) Representative confocal images and quantification of oligodendrocytes (OLIG2+/DAPI+) treated with increasing concentrations of cresol (0, 10, 50 µM) for 24 hr. Scale bars, 20 µm; 10–15 fields (20×) per condition/experiment; *n*=2 independent primary cultures. Data are mean ± S.E.M; *$p < 0.05$, ***$p < 0.001$ based on one-way ANOVA with Dunnett's Multiple Comparison Test; n.s. indicates not significant

cresol could be responsible for the behavioral changes observed in Group I transplant recipients by impacting adult myelination in the mPFC, possibly because this brain region is still capable of generating myelin after development. Several species of Clostridia have been shown to be producers of 4-EPS and cresol (*Persico and Napolioni, 2013*; *Nicholson et al., 2012*), consistent with our findings that the microbiota of transplant recipients displaying an altered behavior (social avoidance and increased despair-like behavior) was enriched with members of the *Lachnospiraceae, Ruminococcaceae*, and other unidentified families within the Clostridiales order. We also detected a disruption in gut biosynthesis of tryptophan, tyrosine, and phenylanine in recipient mice with behavioral changes after transplantation. This might result in changes in the systemic/CNS levels of serotonin and other neurotransmitters, as almost 90% of serotonin production occurs within the gastrointestinal tract from its precursor tryptophan (*Berger et al., 2009*; *Yano et al., 2015*). Intestinal serotonin could cross through the blood brain barrier into the brain to regulate the observed social and despair-like behaviors. Additionally, accumulating evidence suggests that alterations in the glutamatergic system impact the pathophysiology of major depressive disorders (*Tokita et al., 2012*). Interestingly, another metabolite that was significantly downregulated in affected transplant recipients was hexanedioic acid, also known as adipic acid, which can impact glutamate signaling by inhibiting the L-glutamate decarboxylase in the brain (*Wu and Roberts, 1974*). Recent work has demonstrated that epsilon toxin produced by *Clostridium perfringens* Type B is able to bind CNS endothelial cells and white matter tracts, inducing blood brain barrier disruption and oligodendrocyte apoptosis (*Linden et al., 2015*; *Rumah et al., 2013*; *2015*). Although in our studies we could detect *C. perfringens,* its low abundance and the lack of demyelination at the ultrastructural level suggests that other members of the Clostridiales might be driving the behavioral outcome.

In conclusion, our data support the concept that myelinating oligodendrocytes play a pivotal role in the pathogenic process underlying social avoidance, and define the intestinal microbiota as a potential regulator of such behavioral alterations in adult mice.

## Materials and methods

### Animals
Seven-week-old male C57BL/6 and NOD mice were purchased from Jackson Laboratories (Bar Harbor, ME) and housed in specific pathogen-free facilities at Mount Sinai. All procedures were performed in accordance with the Institutional Animal Care and Use Committee guidelines of the Icahn School of Medicine at Mount Sinai (#08–0676, #08–0675; LA10-00398; LA12-00193; LA12-00146).

### Antibiotic treatment
A cocktail consisting of vancomycin (50 mg/kg), neomycin (100 mg/kg), metronidazole (100 mg/kg), and amphotericin B (1 mg/kg) was administered daily by gastric gavage or subcutaneous injection within a volume of 200 µL and 100 µL, respectively. Control mice received dH$_2$O (gastric gavage) or saline (s.c.) as vehicle. Ampicillin (1 g/L) was supplemented in drinking water in the antibiotic-treated group (*Reikvam et al., 2011*). Antibiotics were administered daily for 14 days prior to behavioral testing. During behavioral testing, antibiotics were administered every other day and always after the behavioral tests.

### Behavioral tests
All behavioral tests were recorded and tracked using Ethovision 3.0 (Noldus, Netherlands) for unbiased quantification. Overall anxiety behavior was assessed using Elevated plus maze. Social and despair-like behaviors were assessed using Social interaction and Forced swim tests. To limit carryover effects, behavioral tests were assessed in the order listed below over 14 days. Locomotor activity, Open field, Elevated plus maze, and Social interaction tests were conducted during the first week of testing with 24 hr of recovery between each task, while the Forced swim test was tested the following week.

### Elevated Plus Maze

Mice were placed in the center of the maze, and behavior was recorded for 5 min. Time spent in the open and closed arenas were the dependent variables recorded by video tracking software (Ethovision 3.0, Noldus).

### Social interaction test

A two-stage social interaction test was performed (*Krishnan et al., 2008*). In the first 2.5 min trial, each mouse was allowed to freely explore a square open-field arena (44 × 44 cm) containing a wire cage (10 × 6 cm) on one side. During the second 2.5 min trial (target present), the mouse was reintroduced into this arena now containing a social target (unfamiliar mouse) within the wire cage. Time spent interacting with target or in the corner zones was recorded by video tracking software (Ethovision 3.0, Noldus).

### Forced swim test

Mice were single housed for 24 hr prior to testing and then placed in individual glass cylinders (46 cm height x 18 cm diameter) containing 15 cm of room temperature water. Sessions were videotaped for 6 min and total 'immobility' time was scored blind by a second investigator (*Stratinaki et al., 2013*).

## DNA extraction, 16S rRNA amplification, and multiplex sequencing

All mice used for 16S rRNA sequencing were co-housed per group (3–5 mice per cage) in specific pathogen-free conditions. Fecal pellets were collected directly into sterile 1.5 mL tubes and immediately frozen and stored at -80°C. Cecal content was harvested at the end of each experiment and immediately frozen and stored at -80°C until further analysis. Fecal pellets from co-housed mice were weighted and pooled, and gDNA isolated using Powersoil DNA Isolation kit (Mo Bio Laboratories, Inc., Carlsbad, CA). DNeasy blood and tissue kit (Qiagen, Venlo, Netherlands) was used to isolate gDNA from gut tissue (cecum). For gut microbiome characterization, the V4 hypervariable region of the bacteria 16S rRNA gene was amplified using the universal primers F515 (50-CACGG TCGKCGGCGCCATT-30) and R806 (50-GGACTACHVGGGTWTCTAAT-30). A 12 bp GOLAY error-correcting barcode was added to the reverse primer to enable sample multiplexing. Reactions were performed in triplicate using the AccuPrime Taq DNA Polymerase High Fidelity system (Thermo Fisher Scientific, Waltham, MA). Unless noted, all the analyses were performed using QIIME 1.8.0 as previously described (*Clemente et al., 2015*). Linear discriminant analysis effect size was performed using default parameters (*Segata et al., 2011*). Raw data presented in *Gacias et al. (2016)* (doi:10.5061/dryad.31v06).

## Oligotyping analysis for tissue samples

Oligotype analysis was performed on OTUs belonging to the *Lachnospiraceae, Anaeroplasmataceae*, and *Ruminococcaceae* families, and the Clostridiales order. Singletons and OTUs of low prevalence (<80% of the samples) were removed, and the sequences from the five most abundant OTUs were picked for further analysis. Entropy analysis was performed on this set of sequences to look for highly variable positions within all sequences in each OTU, and the number of oligotypes was chosen based on the entropy peaks generated (*Eren et al., 2014*).

## Bacterial qPCR

qPCR analysis of genomic DNA extracted from tissue of C57BL/6 mice transplanted with microbiota (from vehicle- and antibiotic-treated NOD mice) was performed to quantify the total bacteria, the order of Clostridiales, and the families of *Lachnospiraceae,* and *Ruminococcaceae* in both set of animals. The primer sequences and their features are reported in *Gacias et al. (2016)*. The reaction mixture contained 1× PerfeCTa SYBR Green FastMix, ROX (#101414–278; Quanta Biosciences, Inc., Gaithersburg, MD), 200 nM each primer, 1 µL gDNA in a total volume of 12.5 µL. Each SYBR Green PCR assay was performed in triplicate using the ABI 7900HT Real-Time PCR System (Applied Biosystems of Thermo Fisher Scientific), with the following cycling program: 5 min at 95°C, 30 s at 95°C, 45 s at 55°C/60°C, and 45 s at 72°C for 40 cycles. PCR results were analyzed using RQ Manager software 1.2.2 (Applied Biosystems). The annealing temperature was 55°C for all set of primers apart

from *Ruminococcaceae* (60°C). The genome of *Blautia producta* ATCC 27,340 and *Ruminococcus bromii* ATCC 27,255 were used as reference genomes to construct the standard curves and to calculate the unknown numbers of bacterial gDNA copies in both set of animals as described previous in Tamburini et al. (*Tamburini et al., 2013*).

## RNA isolation and qPCR

Tissue punches were taken from the mPFC or NAc and flash frozen for subsequent processing. RNA was extracted using Trizol (#15596–018; Invitrogen of Thermo Fisher Scientific) and purified with the RNeasy Micro kit (#74004; Qiagen) following the manufacturer's protocol. RNA was reverse transcribed with qScript cDNA Supermix (#95048; Quanta Biosystems, Inc.) and qPCR was performed using Perfecta Sybr Fast Mix Rox 1250 (Quanta Biosystems, Inc.) at the Mount Sinai Shared Resource Facility (primers listed in *Gacias et al., 2016*). Each transcript value was calculated as the average of triplicate samples from several mice per experimental condition (typically 6–12). After normalization to *36b4,* the average value for each transcript was calculated based on the values obtained in all the samples included for each experimental condition.

## RNA Sequencing

RNA from the mPFC was flash frozen for subsequent processing. RNA was extracted using Trizol (Invitrogen), purified with RNeasy Micro kit (Qiagen). RNA was then used for deep sequencing analysis (RNA Seq). Samples were mapped at a rate of 79–80%. After filtering out adaptor and low-quality reads, reads were mapped using TopHat (version 2.0.8) to the mm10 mouse genome (*Trapnell et al., 2009*). The Cufflinks/Cuffdiff suite was used to estimate gene-level expression values as fragments per kilobase of exon model per million mapped fragments and detect differentially expressed genes at a FDR <10% and subjected to Gene Ontology enrichment.

## Mouse primary oligodendrocyte cultures

Primary oligodendrocytes were prepared by sequential immunopanning and kept in undifferentiating conditions as described earlier (*Watkins et al., 2008*) until the onset of experiments. Briefly, oligodendrocyte progenitor cells (OPCs) were isolated from one P6 mouse pup brain using an immunopanning system enabling a purity of 95%. The dissected cortex was chopped in papain buffer, incubated for 20 min at 37°C and titrated in ovomucoid solution (CellSystems GmbH, Troisdorf, Germany). The single cell solution was centrifuged at 1000 rpm for 10 min and resuspended in panning buffer and transferred to a bacterial culture plate coated with Anti-BSL1 *Griffonia simplificonia* lectin (L-1100; Vector Labs, Inc., Burlingame, CA), for negative selection for 15 min, followed by a positive selection step with rat anti-mouse CD140a (10R-CD140AMS; Research Diagnostics, Inc., Flanders, NJ) as primary antibody and AffiniPure goat anti-rat IgG (H+L) (112-005-003; Dianova) as the secondary antibody for 45 min. The supernatant was aspirated, and the positive selection plate was washed with DPBS. The adherent OPCs were removed using trypsin, centrifuged for 10 min at 1000 rpm, resuspended in mouse OPC Sato medium (*Watkins et al., 2008*) and plated in a p100 culture plates coated with poly-d-lysine (P7886; Sigma-Aldrich, St. Louis, MO). The OPCs were cultured in a humidified incubator at 5% $CO_2$ and 37°C with media changes every 2 d. OPCs were maintained proliferating in the presence of bFGF (20 ng/mL) and PDGF (10 ng/mL), while oligodendrocyte differentiation was induced by culturing the cells in the absence of mitogens and adding 60 nM T3 (T5516; Sigma-Aldrich) to Sato medium.

## Cresol treatment

Stock solutions of Cresol (C85751; Sigma-Aldrich) were prepared in DMSO (1000-fold concentrated) and then diluted in differentiation media (SATO+T3) to give final concentrations of 10 µM and 50 µM of Cresol. Primary oligodendrocytes were plated on 0.1 mg/mL poly-d-lysine coated 6-well plates in proliferating conditions (SATO + bFGF and PDGF). Twenty-four hours after plating, cell differentiation was induced by changing the medium to SATO+T3. At this point cells were treated for 24 hr with Cresol at 10 µM or 50 µM as well as DMSO as a control. Cells were gently washed with PBS after completion of the treatment and fixed with 4% paraformaldehyde for 15 min at room temperature for immunocytochemistry experiments.

## Immunohistochemistry and immunocytochemistry

Experimental animals were anesthetized and then perfused with 4% (w/v) paraformaldehyde in 0.1 M phosphate buffer. Whole brains were cryopreserved in 30% (w/v) sucrose, embedded in OCT and sectioned (14 μm). Permeabilization in blocking buffer (PGBA, 10% [v/v] normal goat serum [Vector Laboratories] and 0.5% [v/v] Triton X-100) followed by overnight incubation with primary antibody anti-MBP (clone SMI99, 1:500; BioLegend, San Diego, CA) at 4°C. After incubation with secondary fluorescent antibodies (Donkey anti-mouse Alexa Fluor 594) and nuclear counterstaining with DAPI (1:10,000; Molecular Probes of Thermo Fisher Scientific), immunoreactivity was visualized using LSM780 Meta confocal laser scanning microscope (Carl Zeiss Micro-Imaging, Jena, Germany).

Immunohistochemistry of cultured cells with CNP and OLIG2 antibodies was performed on fixed cells. Cells were grown on CC2-coated 8 well chambers (Lab-Tek, Scotts Valley, CA) for all immunocytochemistry. For staining oligodendrocyte lineage (OLIG2) and differentiation markers (CNP), cells were rinsed gently with PBS and were then fixed with 4% PFA for 15 min at room temperature. Fixed cells were first incubated with blocking/permeabilization solution (PGBA plus 10% normal goat serum, and 0.5% Triton X-100) for 1 hr at room temperature. For co-staining experiments, cells were incubated with additional primary antibodies against OLIG2 (AB9610, 1:1000; Millipore, Darmstadt, Germany) and CNP (SMI91R, 1:500; Covance, Princeton, NJ) overnight at 4°C. One-hour incubation with secondary fluorescent antibodies (Alexa Fluor 594) was performed the following day with counterstaining for DAPI (1:10000) to visualize cell nuclei.

## Image acquisition and quantification

Images were captured with a 20× objective using an LSM 780 Metaconfocal laser scanning microscope (Carl Zeiss MicroImaging, Inc., Jena, Germany). For OLIG2 and CNP cell counts, 10–15 fields were taken per condition. For MBP area quantification, four fields were taken per mouse. Three to four mice were included per treatment condition. MBP+ area and OLIG2+/CNP+ cell counts were quantified using ImageJ (*Liu et al., 2016*; *Rusielewicz et al., 2014*). An unpaired Student's *t* test or one-way ANOVA was performed to assess statistical differences between conditions as indicated in figure legends.

## Electron microscopy

mPFC and NAc samples were prepared from standard electron microscopic analysis as previously described (*Liu et al., 2012*; *2016*). Briefly, mice were transcardially perfused with 0.1 M Millonigs buffer containing 4% paraformaldehyde and 5% glutaraldehyde and post-fixed for 2 wk. Brains were harvested and the region spanning from bregma to 2.5 mm anterior to bregma was vibratome sectioned at 40 μm. Comparable sections ~1.5 mm anterior to bregma and at the level of the forceps (*Liu et al., 2012*) minor of the corpus callosum were selected and embedded in PolyBed resin (Polysciences), thick sectioned (1 μm) and stained with toluidine blue. Using these sections, the mPFC and the core of the NAc were identified, and both regions were thin sectioned (90 nm) and stained with uranyl acetate and lead citrate. For quantitation of myelin thickness, 10 electron micrographs were collected at 10,000× per region using a JEOL JEM 1230EX transmission electron microscope equipped with a Gatan Orius SC1000 side mount CCD camera. Using NIH Image J, the *g*-ratio of a minimum of 100 myelinated axons per region was calculated using the collected electron micrographs.

## Blood glucose measurements

Blood samples were collected by tail snip, and blood glucose was measured using glucose strips (7080G; Bayer Contour).

## Fecal transplantation protocol

At the time of transplantation, microbiota was freshly harvested from the cecum of 8–9-wk-old NOD mice treated with either vehicle or antibiotic. Cecal content was harvested, pooled, homogenized in a 1:4 in sterile solution (1x PBS: 80% glycerol, ratio 1:1), centrifuged at 800 rpm and the supernatant was collected, aliquoted, and stored at -80°C. Recipient 8-wk-old C57BL/6 mice received an oral cocktail of antibiotics (describe above in this section) once daily for 14 consecutive days prior to the transplantation. Recipients were then randomized in two groups (Group I and II), tested for social behavior,

and then immediately started on the re-colonization protocol. To re-colonize the gut of C57BL/6 mice, recipient mice were orally gavaged every other day with 200 µL of cecal content isolated from the vehicle-treated or antibiotic-treated NOD mice over the subsequent 14 d (for a total of 7 times). Behavioral testing was repeated after 15 d of first transplantation, and group-blinded analysis of the results was performed. Cecal and mPFC samples were harvested at the end of the behavioral testing (14 d post-transplantation) and immediately stored at -80°C for further processing and analysis.

### Tissue preparation and metabolomic analysis

Frozen tissues (30 mg) were placed in pre-chilled (-80°C) 2 mL round bottom Eppendorf tubes having a stainless steel ball in it. Next, 400 mL of a pre-chilled (-20°C) mixture of acetonitrile, isopropanol, and deionized water in proportion 3:3:2 (v/v/v) was added. Samples were homogenized using Tissue Lyser (Qiagen) at 25 Hz speed for 5 min. Samples were further centrifuged at 4°C at 12,000 rpm for 3 min. Clean supernatant was transferred into vials or 0.5 mL Eppendorf tubes (to be dried for gas chromatography combined with time-of-flight high-resolution mass spectrometry). Tissue extracts were divided in to three parts: 75 µL for gas chromatography combined with time-of-flight high-resolution mass spectrometry, 150 µL for hydrophilic liquid chromatography coupled with high-resolution mass spectrometry, and 150 µL for hydrophilic interaction chromatography with liquid chromatography and tandem mass-spectrometry. Metabolomic analyses were performed using non-targeted and targeted protocols as previously described (*Tolstikov et al., 2014*; *Urayama et al., 2010*; *Zou and Tolstikov, 2008*). A standard quality control sample containing a mixture of amino and organic acids was injected daily to monitor mass spectrometer response. A pooled quality control sample was obtained by taking an aliquot of the same volume of all samples from the study and injected daily with a batch of analyzed samples and to determine the optimal dilution of the batch samples and to validate metabolite identification and peak integration.

### Metabolite pathway analysis

Identified metabolites were subjected to pathway analysis with MetaboAnalyst 3.0, which consists of an enrichment analysis relying on measured levels of metabolites and pathway topology, and provides visualization of the identified metabolic pathways. Accession numbers of detected metabolites (HMDB, PubChem, and KEGG Identifiers) were generated, manually inspected, and utilized to map the canonical pathways.

### Data processing and statistical analysis

Behavioral and biochemical data were analyzed by unpaired, two-tailed Student's *t* tests or one-way ANOVA followed by Bonferroni post hoc test, as appropriatem using Prism software (GraphPad Software, Inc., La Jolla, CA). Microbiome data were analyzed using QIIME 1.8.0 with default parameters. Statistical significance was assessed using R 3.0.2. Statistical significance for all analyses was accepted at p<0.05. Metabolomic data was analyzed as previously described in Tolstikov et al. (*Tolstikov et al., 2014*).

## Acknowledgements

We thank Dr. Jia Liu (Icahn School of Medicine at Mount Sinai), Dr. Sarah Moyon (Icahn School of Medicine at Mount Sinai), and Dr. Julia Patzig (Icahn School of Medicine at Mount Sinai) for critically reading the manuscript. This work was supported by grant R37NS42925, R01NS52738 and NMSS RG to PC, by the SUCCESS philanthropic grant GCO14-0560 to JCC and by R01NS086444-01 to VZ. Electron microscopy was performed at the VCU - Dept. of Anatomy & Neurobiology Microscopy Facility, supported, in part, by funding from NIH-NINDS Center Core Grant P30 NS047463 and, in part, by funding from NIH-NCI Cancer Center Support Grant P30 CA016059. Computing was partially supported by the Department of Scientific Computing at the Icahn School of Medicine at Mount Sinai.

## Additional information

### Funding

| Funder | Grant reference number | Author |
| --- | --- | --- |
| National Multiple Sclerosis Society | RG 4890 A10/2 | Patrizia Casaccia |
| National Institutes of Health | R37NS42925 | Patrizia Casaccia |
| National Institutes of Health | P30 NS047463 | Jeffrey L Dupree |
| National Institutes of Health | P30 CA016059 | Jeffrey L Dupree |
| National Institutes of Health | R01NS52738 | Patrizia Casaccia |
| National Institutes of Health | R01 NS086444-01 | Venetia Zachariou |
| other | GCO14-0560 | Jose C Clemente |

The funders had no role in study design, data collection and interpretation, or the decision to submit the work for publication.

### Author contributions

MG, JCC, Approved the final version submitted, Conception and design, Acquisition of data, Analysis and interpretation of data, Drafting or revising the article; SG, P-MGS, ST, MA, FZ, NS, VT, MAK, Approved the final version submitted, Acquisition of data, Analysis and interpretation of data, Drafting or revising the article; JLD, Approved the final version submitted, Acquisition of data, Analysis and interpretation of data, Drafting or revising the article, Contributed unpublished essential data or reagents; VZ, Approved the final version submitted, Conception and design, Analysis and interpretation of data, Drafting or revising the article; PC, Approved the final version submitted, Conception and design, Analysis and interpretation of data, Drafting or revising the article, Contributed unpublished essential data or reagents

### Author ORCIDs

Mar Gacias, http://orcid.org/0000-0003-1583-1643
Jose C Clemente, http://orcid.org/0000-0002-3970-9856

### Ethics

Animal experimentation: This study was performed in strict accordance with the recommendations in the Guide for the Care and Use of Laboratory Animals of the National Institutes of Health. All of the animals were handled according to approved institutional animal care and use committee (IACUC) protocols of the Icahn School of Medicine at Mount Sinai (#08-0676, #08-0675; LA10-00398; LA12-00193; LA12-00146).

## Additional files

### Major datasets

The following dataset was generated:

| Author(s) | Year | Dataset title | Dataset URL | Database, license, and accessibility information |
| --- | --- | --- | --- | --- |
| Gacias M, Gaspari S, Mae-Santos P, Andrade M, Zhang F, Shen N, Tolstikov V, Kiebish MA, Zachariou V, Clemente JC, Casaccia P | 2015 | Data from: Microbiota-driven transcriptional changes in prefrontal cortex override genetic differences in social behavior | http://dx.doi.org/10.5061/dryad.31v06 | Available at Dryad Digital Repository under a CC0 Public Domain Dedication |

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
