## [Decision Letter]

Thank you for submitting your work entitled "Microbiota-driven transcriptional changes in prefrontal cortex override genetic differences in social behavior" for consideration by *eLife*. Your article has been reviewed by five peer reviewers, one of whom, Peggy Mason, is a member of our Board of Reviewing Editors and a Senior Editor.

The reviewers have discussed this paper extensively with each another and the Reviewing Editor has drafted this decision to help you prepare a revised submission.

The following individuals involved in review of your submission have agreed to reveal their identity; Jack Gilbert (peer reviewer).

Summary:

There was strong enthusiasm for this work which attempts to tie changes in the gut microbiota to changes in behavior and also changes in myelin-related genes, studied in the prefrontal cortex.

Essential revisions:

1) Ab-gg changes the biota of both strains but only changes NOD's behavior. The authors claim that these data support the idea that a change in gut biota alters behavior only in NODs, so in a strain specific way. An equally plausible interpretation is that a change in gut biota has nothing to do with the change in behavior. So the data are consistent with multiple interpretations.

2) "Throughout the manuscript naming the direction of changes and defining what is normal"/baseline is actively avoided and this causes major confusion. Only differences between treatment groups are reported. This leaves the data floating somewhere in space, completely un-calibrated. Control groups of unmanipulated -in any way – NODs and C57s are needed.

3) The FMT transplant to the ABx-treated C57 rats is reported in such a backhanded way as to completely confuse the reviewers who spent a great deal of time trying to decipher it. This experiment also suffers from a lack of a control – ABx-C57 untreated with gavage (the baseline for this expt).

4) Changes in myelin genes are not changes in myelin. To make the claims that the authors make, some measure of myelination is needed and some degree of specificity to the frontal cortex. And it is not clear why myelin rather than clock genes are highlighted.

Overall there is enthusiasm for the experiments and data and skepticism that the conclusions are as solid as they are presented. Acknowledgement of alternate possibilities is absolutely needed along with some softening of the conclusions.

Additional points:

Reviewer #1:

Why use NOD mice? It think you should be clearer about this when you first mention it at the beginning of the results.

The statement that "ruling out the possibility that the observed behavioralalterations in NOD mice were consequent to a systemic effect of antibiotics." Is not necessarily true right? Maybe I am wrong, but what is gavaged antibiotics interacted with the host cells in the gut, or the immune regulation – to control for this you would need to run germ free models.

It would be nice to run oligotyping on the 16S rRNA analysis of the key Lachnospiraceae, Ruminococcaceae, and Clostridiales strains anyway to explore if there were any strain specific effects that might contribute to the variance in behavioral response. Also comparison of the strain specificity between donor and recipient for the FMT, would significantly advance the case that these were transferred strains, rather than existing strains whose presence was augmented by the FMT community.

Reviewer #2:

While the NOD-derived microbiotal were associated with a number of prokaryotic genotypes and an altered metabolic profile of the gut (e.g. by mass spectroscopy and gas chromotography), no single "responsible" microbe or neurotoxic metabolite was identified. Here, the authors speculate that it may be a "community" effect, but they later discuss also specific metabolite changes (such as 4-EPS/cresol, altered tryptophan/serotonin levels, or hexanedioic acid/ altered glutamate signaling) as possible underlying causes.

While this is not the first study linking gut microbiota in mice with anxiety-like traits (the authors cite five papers and a quick search in Google Scholar adds many more papers), it is the first study to point out secondary changes of myelin gene transcripts in the CNS of antibiotica-treated mice. The same group has previously shown (Nature Neuroscience 2012) that myelination of the cortex can change as a result of social withdrawal, presumably reflecting altered neuronal activity.

I find this work very interesting and conceptually novel in the field. My only concern is that the altered expression of genes implicated in adult myelination should be more carefully discussed. We do not know whether the altered gut microbiom (or any of the metabolites derived) is "directly" causal in changing myelin gene expression of oligodendrocytes, or only "indirectly", i.e. by affecting behaviour first. Indeed, the latter had been shown by the authors' lab to be an important aspect of myelination control.

A decisive experiment would be to compare the RNA profile of oligodendrocytes in prefrontal cortex and white matter tracts, which are less likely to be affected by behavioural changes (e.g. spinal cord).

The authors should also show their immunostainings of myelinated fibers after the transplant. Does it match the strain difference observed before? A nearly 2-fold difference would be visible on Western blots.

The authors should avoid writing "it is not possible to draw a direct link between gut microbial metabolism and mPFC transcriptome". The necessary experiments have just not been carried out.

Reviewer #3:

The sc and gg vehicles produce different baseline behaviors. Is this an effect of the relative stress of the two procedures?

My one big concern is the enormous gastric damage done by the abs. From the first expt it is not clear that the causative agent is the change in microbiota or the stomach inflammation. The one expt that could be used to point to microbiota is the final one where ab-treated C57 mice get either veh or ab-treated NOD mice microbiota. But to get this we need to see the stomachs of the C57 mice in each group. While of course, fashion would have us believe that it is the microbiota, it is formally possible that the cause of the changes is metabolites from the very extreme inflammation. And that the reactions differ in NOD and C57 mice (just as the microbiota differ in these two strains).

Reviewer #4:

The premise of the paper is fascinating and the authors have done an admirable job in synthesizing a connection between gut microbiome and cortical myelin related genes. However, their hypothesis comes undone when making a direct link to behavior.

• I grappled hard and failed to follow the author's logic of the fact that antibiotic administration significantly decreased bacterial diversity of both NOD and C57BL/6 mice and yet the behavioral alterations occurred only in the NOD mice leads to the conclusion that "These findings suggest that NOD treated with vehicle might have a particular microbiota community, sensitive to antibiotic treatment, which could modulate the social and despair-like behaviors". Wouldn't the more parsimonious explanation that the behavior and microbiome be unrelated?

• Indeed the authors prove the opposite of their hypothesis is true in the next few sentences. It looks as rather that C57 mice have some strains enriched that appear to be preventive of enhanced social behavior and mood through antibiotics compared to NOD.

• The authors hypothesis also falls apart at the fecal transplantation stage – when they transplant fecal microbiota from NOD mice, which have decreased despair & increased social activity, it induces the opposite phenotype in antibiotic depleted C57 mice. This is fascinating data but difficult to explain easily. The authors somewhat try to distract the reader from this fact by now calling these groups Group 1 and Group 2.

• Overall, the authors take a rather unexpected point-of-view to interpret the data, which is highlighted by the following statement (among others throughout the manuscript): "Collectively, these findings suggest that the gut microbiota of NOD mice is sufficient to induce a reduction of the social behavior and increase the despair-like behavior independent of the genetic background and mediated through changes in the transcriptomic profile of the mPFC."

In fact, the microbiota of the NOD mice is the baseline and hence cannot be viewed as to "induce" a phenotype. The alternative interpretation, that immediately emerges from the presented data would be: “Collectively, these findings suggest that the ANTIBIOTICALLY-DEPLETED(/MODIFIED) gut microbiota of NOD mice is sufficient to induce an INCREASE of the social behavior and DECREASE in the despair-like behavior independent of the genetic background and mediated through changes in the transcriptomic profile of the mPFC."

And this finding needs to be discussed and explained by the data, i.e. the differences in microbiota and metabolome before and after antibiotic treatment and/or transplantation.

• In their analysis of microbiota in NOD vs. C57Bl6J (under control conditions) it should be made clear, which bacteria are more abundant in NOD mice because it would be important to know which bacteria are depleted in NOD mice that lead to the behavioral performance increase. And this finding needs to be discussed and explained by the data, i.e. the differences in microbiota and metabolome before and after antibiotic treatment and/or transplantation.

• It may be helpful to state what would be expected and then look for deviations from the expectation: Microbe-depleted C57 mice are expected to show a similar microbial profile to the respective donor, i.e. NOD vehicle-treated or NOD antibiotic treated. The authors should provide and discuss evidence that is or isn't the case. Unexpected interpretation happens from this point onwards. Interpretation of this data very much depends on what the baseline is, what is viewed as "normal" and what as "altered".

• It would benefit readability of the manuscript a lot if the author would describe the direction of the change they mean rather than say that behavior was "altered" in many sentences. The same is true for "differences" in the microbiota composition. The use of such terms without mentioning the direction of the change make the findings sound very vague and make it more difficult to interpret the data

• How are the mouse strains different genetically? How could this affect the microbiota present in each strain and subsequently the reaction to the antibiotic? This study needs to take gene-environment interactions into account.

• Why use NOD mice at all, it is a strange choice for behavioral genetic studies; does their diabetes interfere with the interpretation of phenotype?

• The authors’ assertion that social & despair-related behavior is "mediated through changes in the transcriptomic profile of the mPFC" is not proven at all. Correlation dos not mean causation.

• Regarding the myelination data itself the authors show an increase in mRNA and increased intensity of myelin staining. Is intensity staining the best method, does this mean there is more myelination, as in more axons, or are the fibers that are myelinated more heavily myelinated. Looking at structure using electron microscopy would provide an answer. What is the alteration at the level of oligodendrocytes? Have any of the myelin related changes any functional relevance to demyelination-related disorders or to functional myelin responses.

• The Introduction is lacking a coherent flow and would benefit from significant editing. I found the first paragraph of Introduction, whilst interesting, overlong and not really pertinent to study at hand.

• The relevance of current study to schizophrenia & bipolar is not clear and the authors are perhaps conflating this relevance in Introduction. This is further expanded in the Discussion that compares the data with that from an autism model. If this is focused on social behavior, as title but not introduction indicate, then it’s easy to bring all of this together, but currently it is a number of 'characters' in search of a coherent story.

• An example of the lack of flow is going from the statement "Our laboratory has recently demonstrated that mouse models of depression were characterized by down-regulation of oligodendroglial transcripts and myelin thickness in the medial prefrontal cortex (mPFC), thereby supporting a role for myelin in the pathogenesis of depressive disorders” to "Indeed, manipulations of intestinal microbiota have recently emerged as a novel approach to influence brain function and behavioral outcomes" – this is a strange transition.

• The authors state that "Antibiotic regimen was well tolerated" other than bodyweight, which is a relatively crude measure, was there any other change in secretomotor function, defecation rates etc.

• Motor activity needs to be shown at individual time-bins as the overall effect seems to be different in NOD mice if not reaching statistical significant when collapsed overall.

• Is the myelination effects specific to the cortex; Other brain areas such as the amygdala and hippocampus would be equally worthy of investigation and at least addressed.

• The myelination data is intriguing but it’s somewhat strange that the authors chose to follow up on the least-significantly enriched functional category (myelination) in their transcriptome analysis; further scientific justification is needed.

• Were antibiotic treatments continued during the 1.5 weeks behavioral testing?

• y-axis in Figure 4 unclear – what is MBP+ fibers mean pixel intensity? How is this measured?

• Figure 4: it would be worthwile to graphically show the NOD vehicle treated mice for gene expression.

• Results section needs to be cleared up to enhance readability.

• When describing Figure 5, the authors state "characterized by lower levels of myelin gene transcripts […] in Group I and higher in Group II", which is confusing as this was not compared to baseline in C57Bl6. They should state that transcription of these genes was increased in Group II compared to Group I as shown in the figure.

• The color legend in Figure 6 needs to show the Group names (Group I or Group II), to avoid confusion about the interpretation of the data. Also the description of this figure panel states, "significantly distinct clustering of C57BL/6 transplanted mice displaying social avoidance and an increase in despair-like behaviors compared to normal behaving controls" although, according to the data presented in Figure 1, the "normal" behaving animals show more social behavior and less despair-like behavior.

• A list of the differentially expressed genes should be added to supplemental material, together with a list of terms from functional enrichment analysis.

• The Discussion focuses mainly on the metabolomics data, but does not help explaining how the observed phenotypes are connected to a depleted microbiota. Here the authors show that it is the bacteria that remain after antibiotics (and are therefore overtaking/-growing) are necessary for the myelin increase – that's what the fecal transfer shows. if it was the bacteria that are gone who are important for the myelin changes than you would also see the effects in antibiotic C57Bl6 mice. This however, very much depends on the genetic background of the two strains and it's interaction with the microbiome – which is not discussed in this manuscript at all.

• Figure 7 in the last sentence of the Results section should be 7e and this sentence is again misleading, or at least again a rather surprising conclusion similar to the above.

• Not clear why the outcome of the increased myelin gene expression in transplantation of NOD microbiota to C57BL/6 is not shown at the protein level.

• Why are they using two different reference genes in the qPCR data?

• Images don't elude to which part of the medial PFC they imaged. It should be easy enough with the staining to highlight that. What layer of the PFC has more myelin?

Reviewer #5:

1) The notion that the antibiotic treatments alter myelinating cells in the PFC is based on gene expression analysis RT-PCR and histological data showing changes on MBP+ fibers. While RNA levels provides strong evidence for changes in transcription or RNA stability in oligodendrocytes, to conclude that the antibiotic treatment affects myelin, immunostaining for MBP+ fibers is not sufficient. Morphological analysis like the ones the Casaccia group has done in prior studies (e.g. electron microscopy of myelin thickness, quantification of oligodendrocyte and oligodendrocyte progenitor cell density) will be necessary.

2) It would be very important to determine if the changes in myelin gene expression are unique to the PFC or this phenomenon is widespread throughout the brain. The authors should evaluate the expression of the myelin genes affected by antibiotic treatment in a brain region not linked to social/emotional regulation. The interpretation of the findings would be very different if this is a specific or a general phenomenon.

3) There are some confusing aspects regarding the effects of treatments on behavior. For example, in the immobility tests in Figure 1, it seems that the only group with altered behavior is the oral-vehicle group. The behavior of mice treated with oral antibiotics is the same as the one with SI or SI-vehicle (and similar to all C57BL/6). Furthermore, in the measurements of social interactions, it seems that the oral vehicle reduces and the oral-antibiotic increases the time in the interaction zone compared to both groups treated by SI. Difference in behaviors between mice treated by SI vs oral, such as those in the elevated plus maze (Figure 2) leaves me wondering if in some cases lack or presence of effects of antibiotics could be due to a particular susceptibility of a mouse strain to a manipulation such as gavage or injections, and not really to a difference in the changes in microbiota. This issue would be clarified by presenting data on control mice (not treated at all).

4) Did the authors monitor for onset of diabetes in NOD mice (i.e by detecting evidence of glycosuria)? This strain spontaneously develops diabetes in approximately 20-30% of males and the incidence of disease has been linked to changes in the microbiome (See Wen et al. Nature 2008; PMID 18806780). It would be important to know if the behavioral changes with antibiotic treatment in the NOD mice coincide with metabolic disease to determine if there is an interaction among these variables.

5) The authors speculate that neurotoxic metabolites produced by a combination of gut microbia may affect PFC function. This interpretation would be strengthened if these metabolites were measured and detected in the CNS in vehicle and antibiotic treatment mice.

---

## [Author Response]

Essential revisions:

*1) Ab-gg changes the biota of both strains but only changes NOD's behavior. The authors claim that these data support the idea that a change in gut biota alters behavior only in NODs, so in a strain specific way. An equally plausible interpretation is that a change in gut biota has nothing to do with the change in behavior. So the data are consistent with multiple interpretations.*

*2) "Throughout the manuscript naming the direction of changes and defining what is normal"/baseline is actively avoided and this causes major confusion. Only differences between treatment groups are reported. This leaves the data floating somewhere in space, completely un-calibrated. Control groups of unmanipulated -in any way – NODs and C57s are needed.*

*3) The FMT transplant to the ABx-treated C57 rats is reported in such a backhanded way as to completely confuse the reviewers who spent a great deal of time trying to decipher it. This experiment also suffers from a lack of a control – ABx-C57 untreated with gavage (the baseline for this expt).*

We believe that these comments are legitimate, given the fact that we had not clearly explained that we had referred all our analysis to baseline (without explicitly showing the data). We have now modified all the figures to include those data and revised the text accordingly. In order to summarize the overall experimental results, we would like to summarize the results.

We present evidence that untreated NOD (NOD baseline = no phenotype) had no depressive-like phenotype, while NOD mice gavaged with vehicle vehicle showed the behavioral phenotype (NOD vehicle = phenotype), which was not observed when the animals were gavaged with antibiotics (NOD antibiotic gavage = no phenotype). This suggested that the behavioral effect of gavage requires the microbiota of the NOD mice, since it was not observed when it was depleted by the chronic antibiotic treatment. The behavioural effect of daily gavaging observed in NOD was not detected in C57Bl mice, which showed no depressive-like phenotype either untreated or gavaged with vehicle or antibiotic. This indicated that the behavioral effects induced by daily gavaging were dependent both on the specific mouse strain and its microbiota.

We then asked the question of whether transplanting the microbiota from one strain responsive to gavaging (i.e. NOD) to a non-responsive one (i.e. C57) would be sufficient to transfer the behavioural response to daily gavaging. In order to do so, we depleted (using an antibiotic regimen) the microbiota of unmanipulated baseline C57Bl mice and then transplanted them with the microbiota from NOD mice with a depressive-like phenotype (which had been previously induced by vehicle-gavaging). This transplantation was sufficient to induce in the C57Bl recipients the same social avoidance behavior detected in the vehiclegavaged NOD donors. This phenomenon was not observed in C57Bl recipients that received the microbiota of antibiotic-gavaged NOD donors. Overall, these results suggest that the microbiota of vehicle-gavaged NOD mice was sufficient to elicit a depressive-like phenotype when transferred to a different strain of mice with a normal baseline behavior and antibiotically depleted microbiota.

Finally, we sought to identify the bacterial taxa that were associated with this behavioral phenotype. We therefore performed two types of analyses: the first one (presented in the new Figure 3 and associated table of OTUs) focused on differences between vehicle-gavaged NOD (with behavioral phenotype) and untreated NOD baseline mice. This analysis demonstrated the presence of specific taxa in vehicle-gavaged NOD (displaying a phenotype), that were not found at baseline (= no phenotype), and we identify as potential candidates to explain the depressive-like phenotype in vehicle-gavaged NOD mice. In addition, we reasoned that if this behavior was transferable through the microbiota, then the same relevant bacteria present in the NOD donors with the behavioral phenotype should be detected in C57Bl recipients that showed the social avoidance behavior after transplantation. For this reason, we focused on the similarities between the microbiota of vehicle-gavaged NOD donors (which exhibited a depressive phenotype) and C57Bl recipients transplanted with the microbiota of such NOD donors (which showed the depressive-like symptoms after transplantation, but not before). This analysis is presented in the new Figure 6, which identifies members of the Clostridiales, Ruminococcacea and Lachnospiraceae as the main bacterial taxa related to the depressive-like behavior transferred from vehicle-gavaged NOD donors to C57bl recipients. We further validated the enrichment of these taxa in the samples by RT-PCR, confirming the relation of these bacteria with the phenotype.

*4) Changes in myelin genes are not changes in myelin. To make the claims that the authors make, some measure of myelination is needed and some degree of specificity to the frontal cortex. And it is not clear why myelin rather than clock genes are highlighted.*

The reviewer is correct in his/her determination. In the original submission we had included not only transcript levels but also immunohistochemical data. In response to the comment related to regional specificity, we now show that the transcriptional and ultrastructural changes occur in the PFC and not in other related brain regions. The reason we focused on myelin is because our group (Liu et al., Nature Neuroscience 2012; Liu et al., J, Neurosci., 2016) and others (Matinaken, Science, 2012) had previously reported the importance of PFC myelination in mice with depressive-like symptoms.

Reviewer #1:

*Why use NOD mice? It think you should be clearer about this when you first mention it at the beginning of the results.*

Gene-environment interactions are known to affect the development of neuropsychiatric disorders, however the relative impact of these two variables and the potential underlying mechanisms of pathogenesis remain elusive. We simply wanted to use two genetically distinct strains with clear behavioral differences to evaluate the gut microbiota as environmental variable modulating behavior.

*The statement that "ruling out the possibility that the observed behavioralalterations in NOD mice were consequent to a systemic effect of antibiotics." Is not necessarily true right? Maybe I am wrong, but what is gavaged antibiotics interacted with the host cells in the gut, or the immune regulation – to control for this you would need to run germ free models.*

We have rewritten the Discussion to better reflect our results and we have moderated our conclusions.

*It would be nice to run oligotyping on the 16S rRNA analysis of the key Lachnospiraceae, Ruminococcaceae, and Clostridiales strains anyway to explore if there were any strain specific effects that might contribute to the variance in behavioral response. Also comparison of the strain specificity between donor and recipient for the FMT, would significantly advance the case that these were transferred strains, rather than existing strains whose presence was augmented by the FMT community.*

We thank the reviewer for this valuable suggestion. It is indeed possible that some of the differences were attributable to effects at the strain level rather than at the OTU level, and we performed this analysis on all OTUs found to be transferred from the vehicle-gavaged NOD mice to the C57Bl that were not found in the recipients before transplantation (see Figure 6 for a description of this analysis). Results of the oligotyping analysis are presented in Figure 6—figure supplement 3. Most of the OTUs associated with the behavioural phenotype were composed of a single oligotype at high abundance (Figure 6—figure supplement 3, panels A to C and G to O). Three OTUs had two oligotypes with similar distribution of abundance across samples: Blautia producta, an unidentified member of the Lachnospiraceae, and an unidentified member of the Clostridiales. Further inspection of the sequences associated with these oligotypes revealed Blautia producta JCM 1471 as the closest reference sequence in NCBI, while the Clostridiales oligotypes had no close reference sequence. Overall, these results suggest that either a single oligotype or a combination of two oligotypes are dominant within the analyzed OTUs and might drive the observed depressive-like phenotypes.

Reviewer #2

While the NOD-derived microbiotal were associated with a number of prokaryotic genotypes and an altered metabolic profile of the gut (e.g. by mass spectroscopy and gas chromotography), no single "responsible" microbe or neurotoxic metabolite was identified. Here, the authors speculate that it may be a "community" effect, but they later discuss also specific metabolite changes (such as 4-EPS/cresol, altered tryptophan/serotonin levels, or hexanedioic acid/ altered glutamate signaling) as possible underlying causes.

While this is not the first study linking gut microbiota in mice with anxiety-like traits (the authors cite five papers and a quick search in Google Scholar adds many more papers), it is the first study to point out secondary changes of myelin gene transcripts in the CNS of antibiotica-treated mice. The same group has previously shown (Nature Neuroscience 2012) that myelination of the cortex can change as a result of social withdrawal, presumably reflecting altered neuronal activity.

*I find this work very interesting and conceptually novel in the field. My only concern is that the altered expression of genes implicated in adult myelination should be more carefully discussed. We do not know whether the altered gut microbiom (or any of the metabolites derived) is "directly" causal in changing myelin gene expression of oligodendrocytes, or only "indirectly", i.e. by affecting behaviour first. Indeed, the latter had been shown by the authors' lab to be an important aspect of myelination control.*

This is a legitimate comment and for this reason, in this resubmission, we have focused on a careful and detailed analysis of one of the metabolites which was found to be increased in the gut of mice with the behavioral phenotype. In Figure 7, we identify cresol as an important metabolite which was detected at higher levels in the gut of C57Bl recipients transplanted with the microbiota of vehicle-gavaged NOD than in C57Bl transplanted with the microbiota of antibiotic-gavaged NOD and therefore asked whether this metabolite could directly modulate myelin gene expression in cultured oligodendrocytes. In Figure 8 we show that cresol treatment of cultured oligodendrocyte progenitors reduced myelin gene transcripts as it impaired their ability to differentiate. These results are of high relevance and provide a molecular explanation also for the effect of the microbiota on the ultrastructural data in the PFC. Our group and others have previously shown that social avoidance behavior is associated with defective adult myelination in the PFC and the data in this manuscript identify a metabolite, cresol, as a metabolic intermediate with the ability to pass the blood brain barrier and modulate myelin gene expression in oligodendrocyte lineage cells.

*A decisive experiment would be to compare the RNA profile of oligodendrocytes in prefrontal cortex and white matter tracts, which are less likely to be affected by behavioural changes (e.g. spinal cord).*

We did perform this analysis and show that the changes in myelin genes are detected in the prefrontal cortex, but not in other brain regions such as the nucleus accumbens (Figure 5 and supplemental). This suggests that the effect of the microbiota on behavior is consequent to molecular changes occurring in brain regions affecting social behavior, rather than to non-specific changes throughout the brain.

*The authors should also show their immunostainings of myelinated fibers after the transplant. Does it match the strain difference observed before? A nearly 2-fold difference would be visible on Western blots.*

In order to provide a clear evaluation of the effect of transplantation of NOD microbiota to C57Bl recipient mice, we conducted a blinded ultrastructural assessment of the PFC myelination. The new data are now presented in Figure 5 which reveal changes in myelination in the PFC and not in the NAC (Figure 5). These ultrastrictural data are of high relevance, as we and others have previously shown that.

*The authors should avoid to write "it is not possible to draw a direct link between gut microbial metabolism and mPFC transcriptome". The necessary experiments have just not been carried out.*

We have modified the text accordingly.

Reviewer #3:

*The sc and gg vehicles produce different baseline behaviors. Is this an effect of the relative stress of the two procedures?*

As previously mentioned, we believe that this question was consequent to the omission of the untreated baseline group in our original submission. The sc group behaved like the untreated group, only the NOD mice gavaged with vehicle showed a depressive-like behavior, which was not observed when they were gavaged with antibiotics, suggesting that the phenotype was related to a specific microbiota.

*My one big concern is the enormous gastric damage done by the abs. From the first expt it is not clear that the causative agent is the change in microbiota or the stomach inflammation. The one expt that could be used to point to microbiota is the final one where ab-treated C57 mice get either veh or ab-treated NOD mice microbiota. But to get this we need to see the stomachs of the c57 mice in each group.*

We have now included the gross anatomy of the stomachs of these mice to address the reviewer’s concern on gastric damage. While the size of the large intestine was affected by the antibiotic treatment, we did not detect changes in weight, size or presence of any substantial macroscopic damage in the gavaged mice. We also would like to mention that neither the weight nor the blood glucose levels were altered by the procedure.

Reviewer #4:

*The premise of the paper is fascinating and the authors have done an admirable job in synthesizing a connection between gut microbiome and cortical myelin related genes. However, their hypothesis comes undone when making a direct link to behavior.*

[…]

*• It may be helpful to state what would be expected and then look for deviations from the expectation: Microbe-depleted C57 mice are expected to show a similar microbial profile to the respective donor, i.e. NOD vehicle-treated or NOD antibiotic treated. The authors should provide and discuss evidence that is or isn't the case. Unexpected interpretation happens from this point onwards. Interpretation of this data very much depends on what the baseline is, what is viewed as "normal" and what as "altered".*

As discussed above, we believe that the reviewer’s concerns are legitimate since we had not previously explicitly shown the data at baseline. We have now entirely modified the figures, reorganized the text and we hope that the new data and presentation of the manuscript will allow the message to come across clear, solid and consistent. To start with, we would like to clarify the general conclusions drawn from of our results:

(1) NOD baseline = no phenotype

(2) NOD vehicle gavage = behavioral phenotype

(3) NOD antibiotic gavage = no phenotype

(4) C57 baseline = no phenotype

(5) C57 vehicle gavage = no phenotype

(6) C57 antibiotic gavage = no phenotype

At baseline, neither NOD nor C57 animals exhibited a depressive-like behavior (1,4). Gavaging with vehicle induced a phenotype in NOD (2) but not in C57 (5). Gavaging with antibiotics did not result in a phenotype in NOD nor in C57 (3,6). From these findings, we reason that the behavioral phenotype must be related to differences in the microbiota of vehicle-gavaged NOD compared to untreated baseline NOD. Treatment with antibiotics was used as a control to demonstrate that the depressive-like behavior observed in NOD mice requires not only gavaging but also a specific microbiota, as its depletion through antibiotics does not lead to a phenotype. By analyzing bacteria found in vehicle-gavaged NOD mice not present at baseline we identified a first set of taxa potentially responsible for inducing the behavioral changes observed (Figure 3).

We then ask the question of whether this microbiota associated with a social avoidance behavior in NOD mice could be transferred to C57 mice. Our results in this second experiment are as follows:

(7) C57 transplanted with the NOD vehicle gavage = behavioral phenotype

(8) C57 transplanted with the NOD antibiotic gavage = no phenotype

From these results, we examined the OTUs found in vehicle-gavaged animals (phenotype) that were not found at baseline (no phenotype) and that were also found in transplanted C57 animals (7, which also exhibit the behavioral phenotype) but not before transplantation (no phenotype), as shown in Figure 6. We argue that these OTUs are responsible for the depressive-like symptoms observed in donor NOD and recipient C57 mice. Again, we use antibiotics as a control (8) to demonstrate that transplantation from antibiotic-gavaged NOD into C57 does not result in a phenotype. Analysis of the specific OTUs inducing a phenotype identified members of the Clostridiales, Lachnospiraceae, and Ruminococcaceae as candidates for the depressive-like behavior (Figure 6, Figure 6—figure supplement 2 and Figure 6—figure supplement 3).

*• It would benefit readability of the manuscript a lot if the author would describe the direction of the change they mean rather than say that behavior was "altered" in many sentences. The same is true for "differences" in the microbiota composition. The use of such terms without mentioning the direction of the change make the findings sound very vague and make it more difficult to interpret the data.*

We agree with the Reviewer that the initial omission of the baseline data might have resulted in a description of the data that appeared quite convoluted and perhaps not easily accessible. In this revised version, however, the description of the data is accurately depicted in terms of enrichment or depletion relative to baseline levels or by comparing levels before and after transplantation. We sincerely hope that the extensive text and figure revisions have adequately addressed the Reviewer’s concerns.

*• How are the mouse strains different genetically? How could this affect the microbiota present in each strain and subsequently the reaction to the antibiotic? This study needs to take gene-environment interactions into account.*

As the reviewer highlights, the main point of this study is to address gene-environment interactions, not to define the genetic differences between strains, which might entail an entirely different approach and the use of congenic lines. While we cannot exclude the possibility that the genetic information, which is clearly distinct in the two strains, could affect the differences in the microbiota composition, the main message of our study is that microbiota can transfer behavioral and transcriptional traits, regardless of the genotype of the recipient mice.

*• Why use NOD mice at all, it is a strange choice for behavioral genetic studies; does their diabetes interfere with the interpretation of phenotype?*

We would respectfully note that the emphasis of the manuscript was not to study the genetic effect on behavior. As such the selection of two mouse strains that had previously been reported to exhibit behavioural differences (Moy et al., 2008) does not seem “strange” to us, but rather adequate to address the effect of environment/gene interaction. Diabetes develops very late in NOD and to exclude any potential confounder related to metabolic alterations we have included monitoring of blood glucose levels throughout the experimental paradigm. Since no changes in blood glucose levels were detected, we conclude that the NOD were not diabetic and as such is not likely that diabetes could interfere with interpretation of our results.

*• The author's assertion that social & despair-related behavior is "mediated through changes in the transcriptomic profile of the mPFC" is not proven at all. Correlation dos not mean causation.*

We agree with the reviewer and have modified this statement in the text.

• Regarding the myelination data itself the authors show an increase in mRNA and increased intensity of myelin staining. Is intensity staining the best method, does this mean there is more myelination, as in more axons, or are the fibers that are myelinated more heavily myelinated. Looking at structure using electron microscopy would provide an answer. What is the alteration at the level of oligodendrocytes? Have any of the myelin related changes any functional relevance to demyelination-related disorders or to functional myelin responses.

We now include not only transcript levels (measured by RT-PCR), and protein (measured by immunohistochemistry), but also a quantitative ultrastructural analysis and evidence of regional differences in myelination. It is worth mentioning that the selective effect of microbiota in the PFC is not surprising if we consider that this is one region characterized by adult ongoing myelination, and thereby the more likely region to respond to changes in active metabolites like cresol, which have the ability to decrease myelin gene expression in oligodendrocyte progenitors.

*• The Introduction is lacking a coherent flow and would benefit from significant editing. I found the first paragraph of Introduction, whilst interesting, overlong and not really pertinent to study at hand.*

[…]

• An example of the lack of flow is going from the statement "Our laboratory has recently demonstrated that mouse models of depression were characterized by down-regulation of oligodendroglial transcripts and myelin thickness in the medial prefrontal cortex (mPFC), thereby supporting a role for myelin in the pathogenesis of depressive disorders” to "Indeed, manipulations of intestinal microbiota have recently emerged as a novel approach to influence brain function and behavioral outcomes" – this is a strange transition.

The three comments above have been addressed by entirely re-writing the Introduction. We sincerely hope that the revised text will address the reviewer’s comments. In addition, we believe the overall readability of the manuscript has dramatically improved.

*• The authors state that "Antibiotic regimen was well tolerated" other than bodyweight, which is a relatively crude measure, was there any other change in secretomotor function, defecation rates etc.*

We disagree with the reviewer regarding this comment as, even though we have not measured the secremotor function or defection rate, the careful assessment of body weight, body condition (including dehydration), overall stomach appearance and weight, collectively indicate an overall healthy state which would not have been detected in case of severe diarrhea or strong intolerance to the antibiotic regimen.

• Motor activity needs to be shown at individual time-bins as the overall effect seems to be different in NOD mice if not reaching statistical significant when collapsed overall.

We have now included in the revised Figure 1 the measurement of the locomotor activity in NOD and C57BL/6 during the SI test.

• Is the myelination effects specific to the cortex; Other brain areas such as the amygdala and hippocampus would be equally worthy of investigation and at least addressed.

We provide evidence for regional specificity in the PFC. Social behavior is known to be affected by changes in myelination in this area. We have now also included also a detailed comparative transcriptional and ultrastructural analysis at the level of the nucleus accumbens, which was not affected by the manipulation of the microbiota. The issue of regional specificity has been linked to the fact that the PFC is one of the very few regions that displays ongoing active myelination in the adult brain and, therefore, we believe more deeply affected by increased levels of microbiota-produced metabolites, like cresol.

*• The myelination data is intriguing but its somewhat strange that the authors chose to follow up on the least-significantly enriched functional category (myelination) in their transcriptome analysis; further scientific justification is needed.*

We believe that in this revised version we have provided a very convincing argument for the rational behind our interest in the characterization of myelination. We and others had previously described behavioural changes associated with myelination deficits in the PFC. We have also recently shown that enhancement of adult myelination counteracts the effect of social isolation in the induction of social avoidance behavior. All these data and additional rationale is now provided in the Introduction and Discussion of the revised text.

*• Were antibiotic treatments continued during the 1.5 weeks behavioral testing?*

Yes, the antibiotic regimen was continued throughout the behavioral testing period. To clarify this point we have updated the Materials and methods for this section.

*• y-axis in Figure 4 unclear – what is MBP+ fibers mean pixel intensity? How is this measured?*

Using image J we have calculated the amount of NF+ axons that were also immunoreactive for MBP+ staining. This method has been used in other published papers such as Liu J et al. (J Neuroscience 2016) and Rusielewicz T et al. (Glia 2014).

*• Figure 4- it would be worthwile to graphically show the NOD vehicle treated mice for gene expression.*

We have generated new graphs in order to show the vehicle treated group bar instead of a dashed line.

• Results section needs to be cleared up to enhance readability.

The Result section has been re-written to render the data more accessible and to improve flow and Readability.

*• When describing Figure 5, the authors state "characterized by lower levels of myelin gene transcripts […] in Group I and higher in Group II", which is confusing as this was not compared to baseline in C57Bl6. They should state that transcription of these genes was increased in Group II compared to Group I as shown in the figure.*

We believe there might be some confusion in the data interpretation. Group I and Group II refer to microbiota depleted C57BL/6 recipients that received either the microbiota of the vehicle-gavaged NOD (which displayed the social avoidance phenotype, compared to baseline, as shown in the new Figure 1) or that of the antibioticgavaged NOD (which did not display behavioral changes compared to baseline, as shown in Figure 1). The results are now clearly explained in the Result section of the text.

*• The color legend in Figure 6 needs to show the Group names (Group I or Group II), to avoid confusion about the interpretation of the data. Also the description of this figure panel states, "significantly distinct clustering of C57BL/6 transplanted mice displaying social avoidance and an increase in despair-like behaviors compared to normal behaving controls" although, according to the data presented in Figure 1, the "normal" behaving animals show more social behavior and less despair-like behavior.*

We sincerely hope that the extensive revisions have eliminated these points of contention.

*• A list of the differentially expressed genes should be added to supplemental material, together with a list of terms from functional enrichment analysis*

The above mentioned data have been uploaded in Dryad as supplemental table.

*• The Discussion focuses mainly on the metabolomics data, but does not help explaining how the observed phenotypes are connected to a depleted microbiota. Here the authors show that it is the bacteria that remain after antibiotics (and are therefore overtaking/-growing) are necessary for the myelin increase – that's what the fecal transfer shows. if it was the bacteria that are gone who are important for the myelin changes than you would also see the effects in antibiotic C57Bl6 mice. This however, very much depends on the genetic background of the two strains and it's interaction with the microbiome – which is not discussed in this manuscript at all.*

There are two points that are made by the Reviewer. The first one relates to the potential mechanisms linking metabolomics to microbiota and observed phenotype and the second one refers to the possibility that the behavioral changes might be caused by “depletion “rather than “enrichment” for specific taxa. In response to this comment, we now show that cresol, one of the metabolite that we find enriched in the gut of mice with a behavioral phenotype, is capable of impairing myelin gene expression in cultured oligodendrocytes. The transcriptional changes in myelin genes and ultrastructural differences in PFC myelination were only detected in mice with high cresol, which were also characterized by social avoidance behavior and the presence of high levels of Lachnospiraceae, Clostridiales and Ruminococcaceae. The second concept which relates to depletion versus enrichment of specific bacterial taxa has been addressed experimentally in Figure 6 and associated supplement, which clearly shows that the behavioral changes induced by fecal transplantation were detected only in recipients with a high level of bacterial diversity and enrichment in Lachnospiraceae, Clostridiales and Ruminococcaceae.

*• Figure 7 in the last sentence of the Results sections should be 7e and this sentence is again misleading, or at least again a rather surprising conclusion similar to the above.*

The text has been entirely reworded to describe the new changes

*• Not clear why the outcome of the increased myelin gene expression in transplantation of NOD microbiota to C57BL/6 is not shown at the protein level.*

We provide ultrastructural data that allow a much more detailed level of resolution. Since the changes are regional, we do not believe that myelin preparations obtained from the whole brain would have allowed us to detect these regional specific differences.

*• Why are they using two different reference genes in the qPCR data?*

This was a typographical error. We only used the 36b4 as reference gene throughout the analysis, as now clearly indicated in the text.

*• Images don't elude to which part of the medial PFC they imaged. It should be easy enough with the staining to highlight that. What layer of the PFC has more myelin?*

We have now provided a schematic of the region that we have analyzed.

Reviewer #5:

*1) The notion that the antibiotic treatments alter myelinating cells in the PFC is based on gene expression analysis RT-PCR and histological data showing changes on MBP+ fibers. While RNA levels provides strong evidence for changes in transcription or RNA stability in oligodendrocytes, to conclude that the antibiotic treatment affects myelin, immunostaining for MBP+ fibers is not sufficient. Morphological analysis like the ones the Casaccia group has done in prior studies (e.g. electron microscopy of myelin thickness, quantification of oligodendrocyte and oligodendrocyte progenitor cell density) will be necessary.*

In agreement with the Reviewer’s request we have now included electron microscopy of the prefronatal cortex of mice after transplantation.

*2) It would be very important to determine if the changes in myelin gene expression are unique to the PFC or this phenomenon is widespread throughout the brain. The authors should evaluate the expression of the myelin genes affected by antibiotic treatment in a brain region not linked to social/emotional regulation. The interpretation of the findings would be very different if this is a specific or a general phenomenon.*

In agreement with the Reviewer’s request we have now included images and transcriptional analysis of the Nucleus Accumbens and we show that no changes can be detected in this brain region.

*3) There are some confusing aspects regarding the effects of treatments on behavior. For example, in the immobility tests in Figure 1, it seems that the only group with altered behavior is the oral-vehicle group. The behavior of mice treated with oral antibiotics is the same as the one with SI or SI-vehicle (and similar to all C57BL/6). Furthermore, in the measurements of social interactions, it seems that the oral vehicle reduces and the oral-antibiotic increases the time in the interaction zone compared to both groups treated by SI. Difference in behaviors between mice treated by SI vs oral, such as those in the elevated plus maze (Figure 2) leaves me wondering if in some cases lack or presence of effects of antibiotics could be due to a particular susceptibility of a mouse strain to a manipulation such as gavage or injections, and not really to a difference in the changes in microbiota. This issue would be clarified by presenting data on control mice (not treated at all).*

As described before, we agree entirely with the Reviewer. There was confusion that originated from the omission of the control mice, and hope the reviewed manuscript clearly presents our results and conclusions.

*4) Did the authors monitor for onset of diabetes in NOD mice (i.e by detecting evidence of glycosuria)? This strain spontaneously develops diabetes in approximately 20-30% of males and the incidence of disease has been linked to changes in the microbiome (See Wen* et al. *Nature 2008; PMID 18806780). It would be important to know if the behavioral changes with antibiotic treatment in the NOD mice coincide with metabolic disease to determine if there is an interaction among these variables.*

We have monitored glucose levels and body weight in NOD mice and did not detect any changes during the time of our experimentation. The presence of normoglycemia (as shown in Figure 1—figure supplement 3) indicates the absence of diabetes.

*5) The authors speculate that neurotoxic metabolites produced by a combination of gut microbia may affect PFC function. This interpretation would be strengthened if these metabolites were measured and detected in the CNS in vehicle and antibiotic treatment mice.*

This is a good point. However, to perform this experiment would require a substantial financial and time investment that the lab cannot afford at this moment. In order to address the potential direct effect of the highly cell permeable metabolite cresol on the cell of interest (i.e. myelinating cells), we have directly treated cultured oligodendrocytes with cresol and showed that this metabolite has the ability to impair oligodendrocyte differentiation and decrease myelin gene expression.